# Semi-Supervised Object Detection: A Survey on Progress from CNN to Transformer

**DOI:** 10.3390/s26010310

**Published:** 2026-01-03

**Authors:** Tahira Shehzadi, Ifza Ifza, Marcus Liwicki, Didier Stricker, Muhammad Zeshan Afzal

**Affiliations:** 1Department of Computer Science, Technical University of Kaiserslautern, 67663 Kaiserslautern, Germanydidier.stricker@dfki.de (D.S.); 2Mindgarage Lab, Technical University of Kaiserslautern, 67663 Kaiserslautern, Germany; 3German Research Institute for Artificial Intelligence (DFKI), 67663 Kaiserslautern, Germany; 4Department of Computer Science, Electrical and Space Engineering University of Technology, 971 87 Luleå, Sweden; marcus.liwicki@ltu.se

**Keywords:** transformer, object detection, DETR, computer vision, deep neural networks

## Abstract

The impressive advancements in semi-supervised learning have driven researchers to explore its potential in object detection tasks within the field of computer vision. Semi-Supervised Object Detection (SSOD) leverages a combination of a small labeled dataset and a larger, unlabeled dataset. This approach effectively reduces the dependence on large labeled datasets, which are often expensive and time-consuming to obtain. Initially, SSOD models encountered challenges in effectively leveraging unlabeled data and managing noise in generated pseudo-labels for unlabeled data. However, numerous recent advancements have addressed these issues, resulting in substantial improvements in SSOD performance. This paper presents a comprehensive review of 28 cutting-edge developments in SSOD methodologies, from Convolutional Neural Networks (CNNs) to Transformers. We delve into the core components of semi-supervised learning and its integration into object detection frameworks, covering data augmentation techniques, pseudo-labeling strategies, consistency regularization, and adversarial training methods. Furthermore, we conduct a comparative analysis of various SSOD models, evaluating their performance and architectural differences. We aim to ignite further research interest in overcoming existing challenges and exploring new directions in semi-supervised learning for object detection.

## 1. Introduction

Deep learning [1,2,3] has become an active area of research with numerous applications in various fields such as pattern recognition [4,5], data mining [6,7], statistical learning [8,9], computer vision [10,11], and natural language processing [12,13]. It has seen significant achievements particularly in supervised learning contexts, by effectively utilizing a substantial amount of high-quality labeled data. However, these supervised learning approaches [14,15,16], rely on labeled data for training that is costly and time-consuming. Semi-Supervised Object Detection (SSOD) [17,18] bridges this gap by incorporating both labeled and unlabeled data [19]. It shows a significant advancement in the field of computer vision [10,11], particularly for industries where obtaining extensive labeled data [17] is challenging or costly. SSOD is used in various sectors, including Autonomous vehicles [20,21] as well as medical imaging [22,23]. In industries like agriculture [24,25,26] and manufacturing [27], where there is lots of data but labeling is time-consuming, SSOD helps make things more efficient.

Semi-supervised methods [28,29] enhance model performance and reduce labeling needs by employing both unlabeled and labeled data [30,31]. Moreover, previous object detection [32,33] approaches primarily involved manual feature engineering [34,35] and the use of simplistic models. These approaches faced difficulties in accurately identifying objects with different shapes and dimensions [36]. Later, the introduction of Convolutional Neural Networks (CNNs) [37,38] revolutionizes object detection by directly extracting hierarchical features [39] from raw data, enabling end-to-end learning [40] and substantially enhancing accuracy and effectiveness. In recent years, Semi-Supervised Object Detection has made significant improvement, driven by advancements in deep learning architectures [41,42], optimization techniques [43], and dataset augmentation strategies [44,45,46,47]. Researchers have developed various semi-supervised learning (SSL) approaches tailored for object detection, each with distinct strengths and limitations [48,49]. These approaches are mainly categorized into pseudo-labeling [50,51,52] and consistency regularization [53], both of which effectively utilize labeled and unlabeled data during training. Moreover, the integration of SSL methods with state-of-the-art object detection architectures such as FCOS [54], Faster R-CNN [55], and YOLO [56] has significantly enhanced the performance and scalability of Semi-Supervised Object Detection systems. This combination not only improves detection accuracy but also helps models work well with new and unseen datasets.

Object detection has seen remarkable progress with the advent of the DEtection TRansformer (DETR) [57,58,59]. Transformers, originally developed for natural language processing [12,13], excel in capturing long-range dependencies [60] and contextual information [61,62], making them ideal for complex spatial arrangements [63,64] in object detection. Unlike CNNs [38,39,40], which rely on localized convolutions and require non-maximum suppression (NMS) [65] to filter out redundant detections, DETR uses self-attention mechanisms [66,67] and do not need NMS. It considers the object detection task as a direct set prediction problem, eliminating traditional processes like NMS [65] and anchor generation [68]. Despite its advantages, DETR has limitations, such as slow convergence during training and challenges with small object detection. To address these issues, advancements in DETR enhance performance and efficiency through improved attention mechanisms and optimization techniques [69]. Following DETR’s success, researchers are now employing DETR-based networks in Semi-Supervised Object Detection approaches [70,71,72,73,74,75]. This combines DETR’s strengths with semi-supervised learning to use unlabeled data [53], reducing the need for large labeled datasets.

Due to the rapid progress of transformer-based Semi-Supervised Object Detection (SSOD) [19,70] approaches, keeping up with the latest advancements has become increasingly challenging. Therefore, a review of ongoing developments from CNN-based to Transformer-based SSOD methods is essential and would greatly benefit researchers in the field. This paper presents a comprehensive overview of the transition from CNN-based to Transformer-based approaches in Semi-Supervised Object Detection (SSOD). As shown in Figure 1, the survey categorizes SSOD approaches into CNN-based (one-stage and two-stage) [76,77,78,79,80,81,82,83,84,85,86,87,88,89,90,91,92,93,94,95,96,97,98,99] and Transformer-based approaches [73,74,75], highlighting techniques like pseudo-labeling and consistency-based labeling. It also provide details about data augmentation strategies [45,46,47,100,101,102,103], including strong, weak, and hybrid techniques.

Figure 2 depicts a teacher–student architecture tailored for semi-supervised object detection. A pretrained teacher model is utilized to generate pseudo-labels for unlabeled data. These pseudo-labels, along with the labeled data, are then utilized to jointly train the student model. By incorporating pseudo-labeled data, the student model learns from a more extensive and diverse dataset, enhancing its ability to detect objects accurately. Additionally, data augmentation methods are applied to both labeled and pseudo-labeled datasets. This collaborative learning approach effectively leverages both labeled and unlabeled data to improve the overall performance of object detection systems.

Table 1 provides an overview of previous surveys on object detection, highlighting key research in semi-supervised learning. It covers a range of topics from theoretical advancements [104,105] to practical applications [106] across various domains. These surveys investigate diverse methodologies and their effectiveness, including specific applications in tweet sentiment analysis [107] and medical contexts [108]. Recent works explore improvements within machine learning frameworks [109], addressing challenges posed by small data and industrial applications with noisy or incomplete labels [106]. Notably, some surveys focus on deep visual learning and image classification using semi-supervised [48,110], self-supervised [108,111], and unsupervised methods [112], providing valuable insights into their effectiveness and challenges. Collectively, these surveys offer a detailed understanding of the advancements, challenges, and practical implementations in the field of Semi-Supervised Object Detection. While previous surveys have focused on CNN-based SSOD methods, the rise of Transformer-based Semi-Supervised Object Detection requires thorough evaluation to understand their effectiveness and trends.

The remainder of this paper is organized as follows: Section 2, the core of this paper, offers a comprehensive overview of SSOD approaches. Section 3 examines different loss functions used in SSOD. Section 4 presents a comparative analysis of SSOD approaches. Section 5 addresses open challenges and future directions.Section 6 explores the role of SSODs in various vision tasks. Finally, Section 7 concludes the paper.

## 2. Semi Supervised Strategies

Semi-supervised object detection (SSOD) relies on both labeled and unlabeled data to improve detection performance with minimal annotation cost. Existing approaches mainly include one-stage and two-stage methods [71,76,78,79,80,82,96,99], as well as emerging Transformer-based detectors [73,74,75]. Together, these approaches highlight the diverse design spaces and methodological innovations within SSOD. To ensure a representative and comprehensive survey, the papers included in this section were selected based on their relevance to SSOD, publication in reputable venues (e.g., CVPR, ICCV, ECCV, NeurIPS), methodological novelty, and demonstrated empirical impact on the field.

### 2.1. One Stage

#### 2.1.1. One Teacher

One Teacher [99] is a teacher–student framework tailored for one-stage SSOD, particularly optimized for YOLOv5 [117,118]. It addresses key issues in one-stage semi-supervised detection—such as low-quality pseudo labels [50,51,52] and conflicts among multiple detection tasks [119]. To improve training stability and pseudo-label accuracy, One Teacher introduces Multi-view Pseudo-label Refinement (MPR) [120] and Decoupled Semi-supervised Optimization (DSO) [121]. Together, these components reduce noise in pseudo supervision and enable more effective teacher–student learning. This method leverages a single teacher model for semi-supervised learning, primarily tested on COCO 1/5/10% splits for real-time applications. It employs Focal Loss for classification, GIoU Loss for bounding box regression, consistency loss between weak and strong augmentations, and soft pseudo-labeling loss to guide the student model, as illustrated in Figure 3.

#### 2.1.2. DSL

DenSe Learning (DSL) [95] algorithm presents a approach to anchor-free SSOD. As shown in Figure 4, is designed for one stage anchor-free detector like FCOS [54], in contrast to current approaches that mainly concentrate on two stage anchor-based detectors, which are more practical for real-world applications.

DSL addresses key challenges by introducing innovative techniques such as Adaptive Filtering (AF) for precise pseudo-label assignment [91,122], Aggregated Teacher (AT) [123] for enhanced label stability, and uncertainty consistency regularization [124] for improved model generalization. DSL is an anchor-free framework built on FCOS that enhances pseudo-label assignment through adaptive filtering and teacher aggregation. It is evaluated on COCO splits and PASCAL-VOC, employing Focal Loss for classification, IoU/GIoU for regression, uncertainty consistency regularization, and adaptive pseudo-label refinement to improve detection performance.

#### 2.1.3. Dense Teacher

The Dense Teacher [96] framework introduces a innovative approach to Semi-Supervised Object Detection (SSOD) by replacing sparse pseudo-boxes with dense predictions termed Dense Pseudo-Labels (DPL) [125,126], as demonstrated in Figure 5. Post-processing procedures, such as Non-Maximum Suppression [65], are not necessary for this unified pseudo-label [50,51,127] structure. Additionally, a region division strategy is proposed to suppress noise and enhance the focus on key regions, further improving detection accuracy. Overall, Dense Teacher represents a significant advancement in SSOD with its streamlined pipeline and effective utilization of dense pseudo-labels [125,126]. Dense Teacher, compatible with FCOS or RetinaNet, replaces box-level pseudo-labels with full dense predictions, removing the need for NMS. Tested on COCO and VOC, it uses dense pseudo-label supervision, Focal and GIoU losses for the student, and region-division noise suppression to provide richer training signals.

#### 2.1.4. Unbiased Teacher v2

Unbiased Teacher v2 [98] introduces an innovative method that extends the scope of SSOD techniques [78,79,81,87] to anchor-free detectors, alongside the introduction of the Listen2Student mechanism to unsupervised regression loss [78,80] is depicted in Figure 6. Key contributions include expanding the applicability of SSOD to both anchor-based and anchor-free detectors [128], developing a mechanism to address misleading instances in regression pseudo-labels [50,51,127], and reducing performance differences between anchor-free and anchor-based detectors [128] in the semi-supervised domain. An anchor-free and general one-stage method, Unbiased Teacher v2 reduces regression errors using mechanisms like Listen2Student unsupervised regression loss. It also applies Focal Loss for classification, IoU/Smooth-L1 for regression, and weak-strong consistency loss, extending SSOD capabilities to anchor-free detectors on COCO and VOC datasets.

#### 2.1.5. S4OD

S4OD [97], a semi-supervised methodology tailored for one stage detectors, addresses the challenge of extreme class imbalance [129] inherent in these detectors compared to their two stage SSOD [78,79,81]. Shown in Figure 7, S4OD introduces the Dynamic Self-Adaptive Threshold (DSAT) strategy [130]. S4OD dynamically determines pseudo-label selection [50,51,52], balancing label quality and quantity in the classification branch. Additionally, the NMS-UNC module evaluates regression label quality by computing box uncertainties via Non-Maximum Suppression [65], enhancing regression targets [81,82]. S4OD addresses extreme class imbalance in one-stage detectors by dynamically adjusting pseudo-label quality. It is evaluated on COCO and VOC datasets, combining standard Focal and GIoU losses with NMS-based uncertainty loss and dynamic self-adaptive thresholding for pseudo-label selection.

#### 2.1.6. Consistent-Teacher

Inconsistent pseudo labels [50,51,52] in Semi-Supervised Object Detection (SSOD) pose a challenge that Consistent-Teacher [94] addresses. These pseudo labels introduce noise into the student’s training process, which causes serious overfitting [131] problems and compromises the construction of accurate detectors.

As represented in Figure 8, Consistent-Teacher introduces a 3D feature alignment module (FAM- 3D) [132], the Gaussian Mixture Model (GMM), and adaptive anchor assignment (ASA) [118,133] as a strategy to minimize this issue. These components enhance the quality of the pseudo-boxes, dynamically modify the threshold values, and stabilize the pseudo-box matching with anchors. Designed to address inconsistent pseudo-targets, Consistent-Teacher applies 3D feature alignment and adaptive thresholding. On COCO, it uses Gaussian Mixture Model thresholding loss, feature-alignment consistency loss, and adaptive anchor assignment loss to stabilize student learning.

### 2.2. Two Stage

#### 2.2.1. Rethinking Pse

Rethinking Pse [91], as shown in Figure 9, introduces certainty aware pseudo labels [50,51,52] that are specifically designed for object detection. These labels accurately assess the quality of both classification and localization [134], providing a more refined method for generating pseudo labels [50,51,52]. By dynamically adjusting thresholds and reweighting loss functions [135] based on these certainty measurements, this mitigates the challenges posed by class imbalance [129,136,137,138,139]. This method introduces certainty-aware pseudo-labels for two-stage detectors. On COCO and VOC, it uses certainty-aware classification loss, certainty-weighted Smooth-L1 regression, and threshold adaptation loss to jointly model localization and classification confidence.

#### 2.2.2. CSD

CSD [77] (Consistency-based Semi-supervised learning method for object Detection), which utilizes consistency constraints [140] to maximize the use of accessible unlabeled data and improve detection performance, as illustrated in Figure 10. This approach extends beyond object classification to include localization [134], ensuring comprehensive model training [134]. Additionally, this introduces Background Elimination(BE) to lessen the adverse effects of background noise on detection accuracy. CSD focuses on feature-level consistency across augmented views to reduce background noise. Evaluated primarily on VOC and COCO, it employs MSE consistency loss, standard CE and Smooth-L1 losses for labeled data, and a background elimination constraint.

#### 2.2.3. STAC

STAC [78] is a semi-supervised [19,70] framework designed to enhance detection models for visual object recognition using unlabeled data, as shown in Figure 11. The baseline detector employed in the proposed architecture is Faster R-CNN [55]. It follows a two-step procedure where a trained detector is utilized in the first stage to generate high-confidence pseudo-labels [141] from unlabeled images. To ensure consistency and robustness, the model undergoes further training in the second stage using labeled and pseudo-labeled data along with significant data augmentations [46,100]. STAC combines augmentation-driven consistency regularization [142] and self-training [143,144] to extend the state-of-the-art SSL from image classification [48,110] to object detection. A pioneering method for weak-strong augmentation pipelines, STAC uses hard pseudo-label CE loss and Smooth-L1 regression on COCO and VOC, with augmentation-driven consistency to improve student model performance.

#### 2.2.4. Humble Teacher

Humble Teacher [79] proposes semi-supervised approach for contemporary object detectors, utilizing a teacher–student dual model framework, as illustrated in Figure 12. The method incorporates dynamic updates to the teacher model through exponential moving averaging (EMA) [145], employs soft pseudo-labels and multiple region proposals as training targets for the student, and utilizes a detection-specific data ensemble to generate more dependable pseudo-labels. Unlike existing approaches such as STAC [78], which rely on hard labels for sparsely selected pseudo samples, the method leverages soft-labels on multiple proposals, allowing the student to distill richer information from the teacher [78]. Humble Teacher relies on soft pseudo-labels with ensemble-like teacher updates for robust supervision. Tested on COCO and VOC, it uses KL divergence for soft label distillation along with CE and Smooth-L1 for labeled samples.

#### 2.2.5. Combating Noise

The proposal outlined in Combating Noise [84] introduces a method resilient to noise by measuring region uncertainty to mitigate the negative impacts of noisy pseudo-labels [146,147]. With this method, the effects of noisy pseudo-labels are carefully examined, and a metric for measuring region uncertainty is ultimately developed.

By incorporating this metric into the learning framework [148], an uncertainty-aware soft target can be formulated to prevent performance degradation caused by noisy pseudo-labeling [146], as illustrated in Figure 13. Additionally, it mitigates overfitting [131] by allowing multi-peak probability distributions and removing competition among classes. Focused on uncertainty-aware supervision, this method applies region-uncertainty loss and KL divergence for multi-peak distributions on COCO and VOC to handle noisy pseudo-labels.

#### 2.2.6. ISMT

A Semi-Supervised Object Detection technique known as Interactive Self-Training with Mean Teachers (ISMT) [86] introduces an approach to rectify the oversight of inconsistencies among detection outcomes in the same image across various training iterations, as shown in Figure 14. By utilizing non maximum suppression [65] to combine detection outcomes from different iterations and employing multiple detection heads to offer complementary information, this approach boosts the stability and quality of pseudo labels. Moreover, the incorporation of the mean teacher model [145] prevents overfitting [131] and aids in the transfer of knowledge between detection heads. ISMT employs multi-head ensemble distillation and iterative NMS-based pseudo-label correction on COCO. It uses Smooth-L1 regression to refine student predictions while improving supervision reliability.

#### 2.2.7. Instant-Teaching

Instant-Teaching [80] leverages instant pseudo labeling [50,51,52] and extended weak-strong data augmentations [47,103] throughout each training iteration to overcome the limitations of manual annotations in typical supervised object detection frameworks. The system implements Instant-Teaching, a co-rectify approach [87], to improve pseudo annotation quality and reduce confirmation bias [145], as depicted in Figure 15. Instant-Teaching minimizes confirmation bias by generating instant pseudo-labels, applying CE loss, confidence filtering, and co-rectify loss. Its experiments are conducted on COCO datasets.

#### 2.2.8. Soft Teacher

In contrast to earlier multi-stage approaches, Soft Teacher [81] introduces an end-to-end solution for Semi-Supervised Object Detection. The object detection training efficiency is increased by this new framework, which progressively enhances pseudo label [50,51,52] attributes during training [78,149]. As shown in Figure 16, this framework proposes two straightforward yet efficient methods: a box jittering methodology [150] for choosing robust pseudo boxes for box regression learning [151], and a soft teacher mechanism involving classification loss is balanced by the classification score from the teacher network. Soft Teacher emphasizes soft classification guidance and weak-strong consistency with box jittering regression loss on COCO and VOC, providing stable pseudo-supervision for one-stage and two-stage detectors.

#### 2.2.9. Unbiased Teacher

Unbiased Teacher [82] framework tackles the bias issue in pseudo-labeling [50,51,52], prevalent in SSOD due to class imbalances [129,136,137], as shown in Figure 17.

By collaborating to train a student and a teacher, who learns slowly, Unbiased Teacher leverages Exponential Moving Average (EMA) [152] and differential data augmentation [101,102,153] to enhance pseudo-label quality and mitigate overfitting [131]. The approach addresses key challenges in SSOD, including class imbalance and overfitting, leading to notable performance enhancements in object detection. A foundational method in SSOD, it uses balanced classification loss, strong-augmentation consistency, and EMA-based pseudo-supervision on COCO and VOC to improve detection under limited labels.

#### 2.2.10. DTG-SSOD

Using the ‘dense-to-dense’ methodology, Dense teacher Guidance for Semi-Supervised Object Detection (DTG-SSOD) [90] utilizes dense teacher predictions directly to guide student training. As represented in Figure 18, this method is facilitated through techniques such as Inverse NMS Clustering (INC)and Rank Matching (RM) [90], allows the student model to emulate the teacher’s behavior during Non-Maximum Suppression (NMS) [154], thereby receiving dense supervision without relying on sparse pseudo labels. INC clusters candidate boxes similar to the teacher’s NMS process, while RM aligns the score rank of clustered candidates between the teacher and student. This dense teacher-guided approach applies dense teacher guidance loss, rank matching loss, and inverse NMS clustering supervision on COCO datasets, improving pseudo-label quality and student learning.

#### 2.2.11. MUM

MUM [85], a data augmentation approach [101,102,153], is introduced to tackle challenges in effectively utilizing strong data augmentation strategies in SSOD due to potential adverse effects on bounding box localization [103].

As depicted in Figure 19, MUM facilitates mixing and reconstructing feature tiles from mixed image tiles, leveraging interpolation-regularization (IR) [155] for meaningful weak-strong pair generation [156].Unlike traditional SSL methods, MUM allows for the preservation of spatial information crucial for accurate object localization. MUM enforces interpolation regularization and consistency across mixed patches, tested on COCO, to increase student robustness to spatial transformations.

#### 2.2.12. Active Teacher

Iteratively extending the teacher–student structure, the Active Teacher [92] method is used for Semi-Supervised Object Detection (SSOD), as demonstrated in Figure 20. Active Teacher addresses the challenge of data initialization in SSOD by gradually augmenting [45,46,47] the label set through an active sampling strategy, considering factors such as difficulty, information, and diversity of unlabeled examples. Active Teacher significantly enhances the performance of SSOD by maximizing the utility of limited label information and improving the accuracy of pseudo-labels [50,51,52]. Active Teacher introduces active sampling to select informative pseudo-labels, applying weighted CE and pseudo-label regression with EMA updates on COCO.

#### 2.2.13. PseCo

Two essential strategies, pseudo-labeling and consistency training (PseCo) [76], in Semi-Supervised Object Detection (SSOD), highlight the shortcomings of these approaches in terms of efficiently using unlabeled data for learning. Specifically, while existing pseudo labeling [50,51,52] approaches focus solely on classification scores, neglecting the precision of pseudo boxes localization, [134,157] and commonly adopted consistency training methods overlook feature-level consistency crucial for scale invariance. To address these limitations, Noisy Pseudo box Learning (NPL) [146,147] is proposed for robust pseudo label generation and Multi-view Scale-invariant Learning (MSL) [158] is introduced to ensure both label consistency and feature-level consistency, shown in Figure 21.

PseCo combats noisy pseudo-boxes using multi-view scale-invariant consistency loss, alongside CE and Smooth-L1 for supervised data, evaluated on COCO and VOC.

#### 2.2.14. CrossRectify

CrossRectify [87] is a detection framework designed to enhance the accuracy of pseudo labels [50,51,52], by concurrently training two detectors with different initial parameters, as depicted in Figure 22. By utilizing the disparities between the detectors, CrossRectify implements a cross-rectifying mechanism [87] to identify and improve pseudo labels, thereby addressing the inherent constraints of self-labeling [159] techniques. Extensive experiments conducted across 2D [59] and 3D [160] detection datasets validate the efficacy of CrossRectify in surpassing existing Semi-Supervised Object Detection methods. CrossRectify improves pseudo-label reliability through cross-rectification disagreement loss and consistency KL loss on COCO.

#### 2.2.15. Label Match

Label mismatch is tackled from both distribution-level and instance-level perspectives through the Label Match [89] architecture, shown in Figure 23. A re-distribution mean teacher [145] employs adaptive label-distribution-aware [161] confidence criteria for unbiased pseudo-label [162] creation to address distribution-level incompatibilities [81,82]. By incorporating student suggestions into the teacher’s guidance, a proposal self-assignment technique resolves instance-level mismatches stemming [163,164] from label assignment uncertainty.

Furthermore, the utilization of a reliable pseudo label mining technique [165] enhances efficiency by converting ambiguous pseudo-labels into dependable ones. Label Match leverages adaptive label-distribution-aware loss, proposal self-assignment, and reliable pseudo-label mining to improve semi-supervised training on COCO.

#### 2.2.16. ACRST

Adaptive class-rebalancing self-training, or ACRST [83], as illustrated in Figure 24, introduces a new memory module called CropBank to address the major problem of class imbalance [136,137] in SSOD. In SSOD, class imbalance [138,139], especially foreground-background and foreground-foreground imbalances—presents serious difficulties that impact the quality of pseudo-labels [50,51,52] and the performance of resulting models. By incorporating foreground examples from the CropBank, ACRST dynamically rebalances the training data, thereby reducing the effects of class imbalance. ACRST addresses class imbalance with foreground–background and foreground–foreground rebalancing, along with two-stage pseudo-label filtering loss on COCO and VOC.

Additionally, to tackle the issue of noisy pseudo-labels [146,147] in SSOD, a two-stage filtering technique [166] is suggested to produce accurate pseudo-labels.

#### 2.2.17. SED

An innovative method called Scale-Equivalent Distillation (SED) [88] introduces an end-to-end knowledge distillation framework [167] that is both straightforward and efficient. SED diminishes noise from erroneous negative data, enhances localization accuracy, and deals with high object size variance by enforcing scale consistency regularization [124], as represented in Figure 25.

Furthermore, a re-weighting technique [168] effectively minimizes class imbalance [136,137,138,139] by implicitly identifying potential foreground areas from unlabeled data. SED applies scale consistency regularization and cross-scale distillation, with re-weighted classification loss, on COCO datasets to improve multi-scale detection.

#### 2.2.18. SCMT

The objective of Self-Correction Mean Teacher(SCMT) [93] is to reduce the negative impact of noise present in pseudo-labels [50,51,52] by dynamically modifying loss weights for box candidates. Depicted in Figure 26, SCMT effectively prioritizes more reliable box candidates during training by utilizing confidence scores derived from both localization accuracy [134] and classification scores. This novel approach outperforms existing methods [78,79,82], demonstrating its potential to improve the performance of object detection models in real-world contexts. SCMT incorporates self-correction weighting and confidence-based localization weighting alongside standard R-CNN losses, tested on COCO, for robust student training.

### 2.3. End to End

#### 2.3.1. Omni-DETR

In order to improve detection accuracy while lowering annotation costs, the Omni-DETR [75] framework is shown in Figure 27, incorporates a variety of weak annotations [169], including picture tags, item counts, and points.

By integrating recent developments in end-to-end transformer-based detection architecture [170,171] and student-teacher-based Semi-Supervised Object Detection [78,82], Omni-DETR enables the use of unlabeled and poorly labeled data to produce precise pseudo labels [50,51,52]. Omni-DETR integrates DETR-style bipartite matching with pseudo-label filtering for both COCO and weakly annotated datasets, using CE and GIoU for labeled data.

#### 2.3.2. Semi-DETR

Semi-DETR [73] employs a Stage-wise Hybrid Matching strategy [172] to combine one-to-one [74] and one-to-many [173] assignment strategies, enhancing training efficiency and providing high-quality pseudo-labels. [50,51,52]. As represented in Figure 28, a Cross-view Query Consistency method [174] eliminates the need for deterministic query correspondence, facilitating the learning of semantic feature invariance. Additionally, the Cost-based Pseudo Label Mining [165] module dynamically identifies reliable pseudo boxes for consistency learning. Semi-DETR employs hybrid matching (one-to-one and one-to-many), cross-view query consistency, and cost-based pseudo-label mining on COCO to improve semi-supervised learning.

#### 2.3.3. Sparse Semi-DETR

Sparse Semi-DETR [74], an end-to-end Semi-Supervised Object Detection system based on transformers. This solution deals with problems regarding the quality of object queries in particular and resolves them. Training efficiency is slowed and model performance is gets worse by inaccurate pseudo-labels [75] and redundant predictions, especially for tiny or obscured objects. As illustrated in Figure 29, to improve object query quality and greatly increase detection capabilities for tiny and partially obscured objects.

Sparse Semi-DETR includes a Query Refinement Module [175]. Robust pseudo-label filtering modules further improve detection accuracy and consistency by filtering only high-quality pseudo-labels [80,81]. Sparse Semi-DETR refines queries and filters unreliable pseudo-labels using Smooth-L1/GIoU losses on COCO, improving transformer-based detection.

#### 2.3.4. STEP-DETR

STEP-DETR [176] introduces a new paradigm for transformer-based semi-supervised object detection by integrating a Super Teacher model with pseudo-label guided text queries to enhance DETR’s reasoning and robustness. As shown in Figure 30, STEP-DETR [176] enriches the detection process by converting high-confidence pseudo-labels [50,51,52] into textual descriptions that serve as semantic prompts for the detector’s query embeddings. This cross-modal guidance enables the model to better align object queries [175] with meaningful semantic cues, significantly reducing ambiguity in query-to-object matching. The Super Teacher provides high-quality pseudo-labels through multi-scale and multi-augmentation fusion, which are then transformed into structured text prompts to refine the student model’s query initialization and attention patterns. STEP-DETR introduces text-guided query alignment, supervised DETR losses, and Super Teacher pseudo-label filtering for COCO and weak supervision, enabling cross-modal semi-supervised detection.

## 3. Loss Function

In semi-supervised object detection (SSOD), various loss functions are used to handle labeled and unlabeled data effectively. Below, we provide both the mathematical formulations and the specific roles of each loss.

### 3.1. Smooth L1 Loss

Smooth L1 loss [89,177,178] is widely used for bounding box regression due to its robustness to outliers and noisy annotations. It applies a quadratic penalty to small errors and a linear penalty to large errors. In SSOD, Smooth L1 is applied to both ground-truth boxes and teacher-generated pseudo-boxes, making it suitable for supervising the student model even when pseudo-labels contain localization noise.$$L_{\mathrm{Smooth}\;L1}\left(x\right)=\left\{\begin{array}{cc} 0.5\;x^{2}, & \mathrm{if}\;|x|<1,\\ |x|-0.5, & \mathrm{otherwise}. \end{array}\right.$$

### 3.2. Distillation Loss

The transfer of Knowledge [179] from a teacher model based on labeled data to a student model with utilization of unlabeled samples is facilitated by distillation loss [79,180]. Distillation loss can be shown as:$$L_{\mathrm{Distill}}=\tau^{2}\;\mathrm{KL}\left(\sigma \left(\frac{z_{T}}{\tau }\right)\;∥\;\sigma \left(\frac{z_{S}}{\tau }\right)\right)$$
where τ is the temperature, and $z_{T}$, $z_{S}$ are teacher and student logits. In SSOD, the teacher produces predictions for unlabeled images, and the student learns to imitate these predictions. This allows the detector to benefit from unlabeled data without explicit annotations.

### 3.3. Focal Loss

Focal Loss [181] addresses the severe foreground–background imbalance commonly observed in object detection [129,136,137,138] by reducing the contribution of easy negative samples during training. It is defined as:$$L_{\mathrm{Focal}}=-\alpha_{t}{\left(1-p_{t}\right)}^{\gamma }\log\left(p_{t}\right),$$
where $p_{t}$ denotes the predicted probability for the ground-truth class, $\alpha_{t}$ is a weighting factor, and γ is the focusing parameter that controls how strongly easy examples are down-weighted. In the context of SSOD, focal loss is particularly beneficial because pseudo-labels often include many easy background predictions. By suppressing their influence, focal loss prevents these trivial samples from dominating the optimization process, thereby stabilizing training and improving robustness to pseudo-label noise.

### 3.4. KL Divergence

Using in semi-supervised scenarios [19,28,70] to align predictions made on labeled and unlabeled data, KL divergence loss [79,84,182,183] minimizes the difference between probability distributions. The KL divergence between distributions *P* and *Q* is:$$\mathrm{KL}\left(P∥Q\right)=\sum_{i}P\left(i\right)\log\frac{P(i)}{Q(i)}$$Consistency-based SSOD methods often minimize KL divergence between predictions obtained under weak and strong augmentations of the same unlabeled image, encouraging stable and coherent estimator behavior.

### 3.5. Quality Focal Loss

Quality Focal Loss (QFL) [96,184] jointly models the classification probability and the localization quality. Its formulation is:$$L_{\mathrm{QFL}}=-{\left|q-p\right|}^{\beta }\left(q\log(p)+(1-q)\log(1-p)\right)$$In SSOD, pseudo-boxes may vary widely in quality. QFL naturally reduces the impact of low-quality pseudo-labels while emphasizing reliable teacher predictions.

### 3.6. Consistency Regularization Loss

The loss of regularization consistency [77,88] ensures consistency in predictions across different views of the same input data, enhancing model robustness and generalization in SSOD. It penalizes inconsistencies, prompting the model to learn invariant features [158], thereby improving performance across varied datasets.A common form is Mean Squared Error (MSE) consistency:$$L_{\mathrm{Cons}}={∥f\left(x\right)-f\left(\tilde{x}\right)∥}_{2}^{2}$$

### 3.7. Jensen–Shannon Divergence

Jensen–Shannon divergence [185,186] symmetrically measures distribution similarity. It is defined as: $$\mathrm{JSD}\left(P∥Q\right)=\frac{1}{2}\mathrm{KL}\left(P∥M\right)+\frac{1}{2}\mathrm{KL}\left(Q∥M\right),\;M=\frac{1}{2}\left(P+Q\right)$$Some SSOD methods regularize predictions by minimizing JSD across predictions from multiple views or across ensemble teachers, helping avoid overconfident or inconsistent pseudo-labels.

### 3.8. Pseudo-Labeling Loss

Pseudo-Labeling Loss [187] enables semi-supervised learning [19,28,70] by generating labels for unlabeled samples using model predictions. The loss is applied only when the model is sufficiently confident, ensuring reliable supervision from pseudo-labels:$$L_{\mathrm{PL}}=-⊮\left(\max\left(p\right)>\tau \right)\sum_{c}{\hat{y}}_{c}\log p_{c}$$
where $\hat{y}$ is the pseudo-label and τ is the confidence threshold. This strategy is central to teacher–student SSOD frameworks, enabling the student model to learn effectively from high-confidence predictions on unlabeled data.

### 3.9. Cross-Entropy Loss

The difference between the estimated probability distribution and the actual distribution of labels is measured by the Cross-Entropy Loss [78,82,188]. By encouraging the model to reduce the gap between the ground truth and the predicted probabilities, this loss increases the classification accuracy. Its formulation is:$$L_{\mathrm{CE}}=-\sum_{c}y_{c}\log p_{c}$$In SSOD, CE provides strong supervision for labeled data and acts as an anchor that stabilizes training when combined with pseudo-label- or consistency-based objectives.

## 4. Datasets and Comparison

In the object detection, having challenging datasets is crucial to ensuring fair and accurate evaluations of different algorithms.

### 4.1. Datasets

Publicly available object detection datasets such as MS-COCO and PASCAL-VOC have become the foundation for most SSOD benchmarks. However, their use in semi-supervised settings differs from conventional supervised tasks. Typically, a small percentage (1%, 5%, or 10%) of the dataset is treated as labeled, while the remaining images are used as unlabeled data. This split simulates real-world conditions where labeled data are scarce and costly to obtain. The Microsoft Common Objects in Context (MS-COCO) dataset contains approximately 118,000 training images and 5000 validation images across 80 object categories. For SSOD experiments, researchers often select a subset of labeled samples and use the remaining unlabeled images to explore model performance under limited supervision. MS-COCO’s large scale, object diversity, and scene complexity make it a challenging benchmark for semi-supervised learning, testing both model robustness and generalization. The PASCAL Visual Object Classes (VOC) dataset, consisting of around 20 object categories, offers simpler but well-structured images for evaluating detection performance. SSOD studies often combine VOC 2007 and VOC 2012 for training, splitting the data into small labeled subsets (e.g., 10% or 20%) and using the rest as unlabeled data. This setup allows researchers to test whether semi-supervised techniques can maintain high performance with limited supervision.

In SSOD frameworks, the labeled data provide initial supervision to train a base detector, while unlabeled data are incorporated through pseudo-label generation and consistency regularization. The teacher model generates high-confidence pseudo-labels for the unlabeled set, which the student model uses for joint training. Data augmentation, both weak and strong, is crucial in this process to improve generalization and mitigate label noise. Therefore, these datasets are not only benchmarks but also active participants in the semi-supervised learning pipeline.

### 4.2. Comparison

The performance of object detection methods has been extensively evaluated on benchmark datasets such as COCO and PASCAL. These evaluations show the progress and effectiveness of both one-stage and two-stage detection approaches, as well as end-to-end methods, in improving detection accuracy over various training epochs. Table 2 offers the performance comparison of various methods on COCO dataset [189]. One stage methods, including One Teacher [99], DSL [95], Dense Teacher [96], demonstrate incremental improvements with increasing training epochs. As illustrated in Figure 31, subfigures (a)–(c) present a comparative visualization of these approaches on the COCO dataset. A clear upward trend is observed, where performance consistently improves from one-stage CNN-based methods to two-stage and finally to transformer-based end-to-end architectures. This confirms that transformer-based SSOD models leverage unlabeled data more effectively and achieve higher mAP with fewer labeled samples.

Two stage methods, such as Rethinking pse [91], STAC [78], and Combating Noise [84], exhibit consistent enhancement in performance metrics over epochs. Notably, DETR-based models like Omni-DETR [75] and Semi-DETR [73] showcase significant performance gains, highlighting the effectiveness of Semi-Supervised Object Detection strategies, as shown in Figure 31. The visual comparison in Figure 31c further demonstrates how transformer-based methods capture long-range dependencies and improve pseudo-label precision, resulting in stronger generalization than CNN-based detectors.

Table 3 shows the performance metrics of various object detection methods across different stages on the PASCAL dataset [190]. In the One stage, methods like S4OD [97], Dense Teacher [96], DSL [95] exhibit competitive performance in terms of AP50, AP50.95, and AP75 scores. Two-stage methods like Soft Teacher [81], Combating Noise [84], and Instant-Teaching [80] display significant variations in performance across different metrics.

Finally, end-to-end methods like Semi-DETR [73] and Sparse Semi-DETR [74] showcase significant performance, indicating the efficacy of SSOD approaches, as illustrated in Figure 32. Figure 32 provides a similar comparison on the PASCAL-VOC dataset, confirming that transformer-based SSOD models consistently outperform CNN-based counterparts even when trained with a small fraction of labeled data. The figure highlights the same progression pattern observed in COCO—demonstrating the scalability and robustness of modern end-to-end SSOD frameworks.

Overall, Figure 31 and Figure 32 emphasize a clear research trend: Semi-Supervised Object Detection has evolved from conventional CNN-based architectures toward more efficient and accurate transformer-based models.

## 5. Open Challenges & Future Directions

Although Semi-Supervised Object Detection (SSOD) has progressed rapidly, from early CNN-based pipelines to sophisticated transformer-driven architectures, this evolution reflects more than just improved accuracy or architectural complexity. A deeper examination reveals an important conceptual shift in how the field interprets uncertainty, supervision, and the role of unlabeled data, as summarized in Table 4, which provides a comparative overview of the strengths and limitations of existing semi-supervised object detection methods.

CNN-based SSOD methods traditionally treated pseudo-labels as unreliable approximations of ground truth. As a result, much of the research effort centered on preventing the student from overfitting to noisy teacher predictions through threshold tuning, uncertainty modeling, and ensemble-based refinement. These strategies expose a fundamental limitation: CNN detectors rely heavily on local heuristics such as anchor assignment, IoU thresholds, and NMS rules—elements that amplify pseudo-label errors in semi-supervised settings.

Transformer-based SSOD approaches offer a different paradigm. Their global attention mechanisms reduce reliance on these brittle heuristics, shifting the focus from correcting pseudo-labels to interpreting and leveraging them. Methods such as Semi-DETR, Sparse Semi-DETR, and STEP-DETR reimagine pseudo-labels as semantic cues that guide query refinement, cross-view consistency, and even text-based reasoning. This marks a conceptual turning point: SSOD is evolving from pseudo-label cleaning toward representation-level alignment and high-level semantic guidance. Yet this progress introduces new challenges that must be addressed for SSOD to mature into a practical, deployable technology. The following subsections outline the major open issues and potential research directions.

**System Complexity and Deployment:** Many state-of-the-art SSOD frameworks rely on multi-stage pipelines with teacher–student models, pseudo-labeling, and consistency regularization. These components improve accuracy but increase computational cost and memory usage, making real-time deployment difficult. Transformer-based SSOD models, in particular, struggle with dense attention mechanisms and iterative optimization. Lightweight architectures, model pruning, quantization, and distributed training are promising directions for reducing system complexity. Future work must explore model compression, distillation, quantization, and efficient transformer variants to balance performance with practicality. Designing lightweight SSOD systems without sacrificing robustness remains an open research front.

**Maintaining Accuracy and Robustness:** Noisy pseudo-labels, domain shifts, and real-world conditions such as occlusion or class imbalance can degrade performance. Techniques like adaptive pseudo-label reweighting, uncertainty-aware learning, and online teacher updates can help maintain accuracy. Continuous and domain-adaptive learning strategies further support robust generalization to unseen data. A key challenge is developing models that not only detect noise in pseudo-labels but actively correct and learn from it, closing the loop between representation learning and uncertainty estimation.

**Evaluation and Benchmarking:** Existing benchmarks such as COCO and PASCAL-VOC often fail to capture realistic deployment challenges. Future research should establish datasets with domain diversity, incremental annotation, and real-world constraints. Standardized evaluation protocols and reproducible pipelines are essential for fair comparison across SSOD methods. Future benchmarks should introduce mixed-quality annotations, temporal or streaming data and cross-domain unlabeled pools. Standardizing evaluation pipelines and reporting practices will help ensure fair comparisons and improve reproducibility.

**Scalability and Reproducibility:** The growing complexity of SSOD architectures makes training resource-intensive and hyperparameter tuning challenging. Open-source frameworks, transparent reporting, and standardized evaluation settings are crucial to ensure reproducibility and wider accessibility. To improve scalability and reproducibility, the field needs more transparent reporting of experiment settings, robust open-source implementations, standardized SSOD training recipes and automated hyperparameter tuning or adaptive schedules. Without addressing reproducibility, many SSOD innovations risk becoming difficult to validate or build upon.

**Domain Adaptation and Hybrid Methods:** Improving generalization through domain adaptation and transfer learning is key for real-world deployment. Hybrid approaches combining semi-supervised, self-supervised, and transfer learning, alongside model compression, can enhance both efficiency and detection performance. Hybrid approaches that blend semi-supervised learning, self-supervised learning, and domain adaptation appear promising. Cross-modal supervision as seen in STEP-DETR’s text-guided queries may further enhance generalization.

SSOD has moved beyond simple pseudo-label refinement toward a more nuanced understanding of representation learning, semantic guidance, and uncertainty modeling. However, limitations in computational efficiency, robustness, benchmark realism, and training stability continue to hinder widespread adoption. Addressing these open challenges will determine whether SSOD advances from a promising research direction into a reliable, scalable technology suitable for industry-level applications.

## 6. Applications

### 6.1. Image Classification

Semi-supervised learning has significantly advanced image classification [22,23], especially in domains with limited labeled data [191]. In medical imaging [192,193,194], SSOD enables accurate disease diagnosis from X-rays and MRIs even with few labeled examples [195]. Similarly, remote sensing [196,197] benefits from improved classification of land cover and environmental changes, aiding urban planning and disaster management. In autonomous vehicles [198,199], it enhances object and pedestrian classification, promoting safer navigation. The primary challenge SSOD addresses in image classification is the scarcity and imbalance of labeled data across diverse visual categories. By leveraging unlabeled images, it reduces annotation costs while maintaining robustness and generalization under domain shifts. Architecturally, effective SSOD models incorporate adaptive feature extractors, teacher–student frameworks, and noise-tolerant pseudo-labeling pipelines capable of handling heterogeneous image sources and visual variations. Techniques like consistency regularization [124] and pseudo-labeling [50,51,52] are critical in stabilizing training and improving accuracy.

### 6.2. Document Analysis

SSOD is increasingly applied to document analysis [200,201,202,203], efficiently detecting and classifying text blocks, tables, and images [204,205,206,207]. In legal, financial, and academic contexts, where large volumes of documents must be processed [208,209,210,211], SSOD reduces reliance on extensive labeled datasets. The key challenges here include variability in document layouts, fonts, and noise from scanning or handwriting. SSOD addresses these by learning structure-invariant representations through multi-scale attention and contextual feature aggregation. Architecturally, systems often combine text-visual fusion modules with region-proposal refinement layers, ensuring semantic consistency across labeled and unlabeled samples. Methods like self-training [212] and consistency-based regularization [124] improve detection robustness under diverse document formats.

### 6.3. Three-Dimensional Object Detection

In 3D object detection [213,214], SSOD improves accuracy and robustness by leveraging both labeled and unlabeled point cloud or multi-modal data. For autonomous driving [20,21], it enhances detection of pedestrians, vehicles, and obstacles using LIDAR and camera inputs [215,216,217,218]. In robotics, it supports precise manipulation and obstacle avoidance, while in AR/VR, it ensures accurate spatial integration of virtual elements with real-world environments. Challenges include sparse or incomplete 3D data, dynamic environments, and cross-modal inconsistencies. Architecturally, effective solutions use cross-modal consistency modules, voxel-based pseudo-label refinement, and memory-efficient 3D feature backbones to enable real-time performance while maintaining robustness under partial observations.

### 6.4. Network Traffic Classification

SSOD is effective for network traffic classification [219,220], identifying and categorizing traffic patterns even with limited labeled data [221]. This is crucial for detecting anomalies and security threats while maintaining network performance. The main challenge is the high volume, heterogeneity, and evolving nature of network traffic. SSOD addresses these issues by exploiting unlabeled traffic data to improve detection of malicious activities [222,223]. Architecturally, models integrate temporal pattern learning, feature embedding regularization, and adaptive pseudo-labeling to maintain robustness across changing network conditions.

### 6.5. Speech Recognition

In speech recognition [224,225,226,227], SSOD improves transcription and phoneme classification even with limited labeled audio. It enhances the separation of speech from background noise and adapts to diverse linguistic and acoustic conditions [228,229,230,231]. Challenges include speaker variability, noisy environments, and limited labeled corpora. SSOD addresses these via cross-modal consistency modules, pseudo-label refinement, and memory-efficient feature extraction, enabling real-time transcription and scalable deployment in voice-controlled systems.

### 6.6. Drug Discovery and Bioinformatics

SSOD accelerates drug discovery [232,233] and bioinformatics tasks [193,234] by improving identification and classification of molecular structures [235,236] and biological entities [237,238]. It reduces reliance on scarce labeled molecular data while handling high-dimensional biological features. The challenge is the complexity and heterogeneity of biochemical data. SSOD addresses this by learning latent molecular representations that generalize across molecular variations. Architecturally, effective models use graph-based detection networks, uncertainty calibration layers, and transfer learning mechanisms [179] to integrate heterogeneous datasets, enabling scalable and interpretable molecular detection for precision medicine.

## 7. Conclusions

This survey presented a comprehensive overview of Semi-Supervised Object Detection (SSOD) by examining both CNN-based and Transformer-based approaches within a unified framework. The primary contribution of this work is its explicit effort to bridge these two methodological lines, which are often treated separately in existing literature. By analyzing their architectural characteristics, learning strategies, and pseudo-labeling mechanisms in parallel, the survey highlights the continuity and differences between CNN-driven and Transformer-driven designs. A second major contribution is the introduction of a structured comparative perspective through a consistent taxonomy and benchmark-oriented organization. This structure enables clearer and more systematic comparison across SSOD methods, offering a coherent view of how architectural developments shape the handling of unlabeled data. Through this, the survey identifies common patterns, gaps, and limitations in current approaches, outlining the areas where further research is needed. Overall, the shift from CNN-based to Transformer-based designs marks an important transition in SSOD. The analysis in this survey provides a consolidated reference for ongoing developments and future research directions in semi-supervised object detection. 

## Figures and Tables

**Figure 1 sensors-26-00310-f001:**
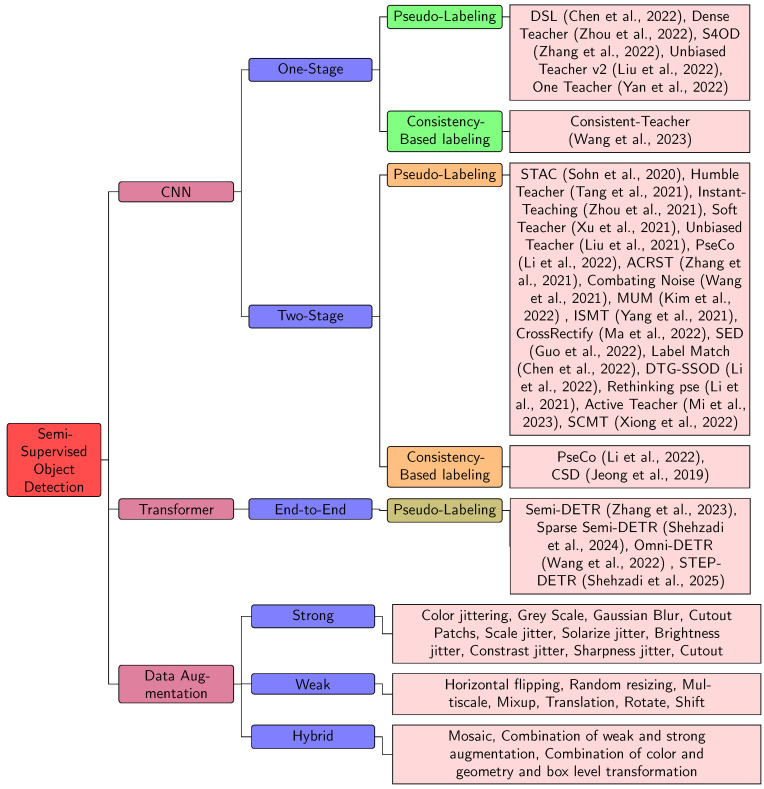
Semi-Supervised Object Detection: A Comprehensive Review and Taxonomy of Techniques.

**Figure 2 sensors-26-00310-f002:**
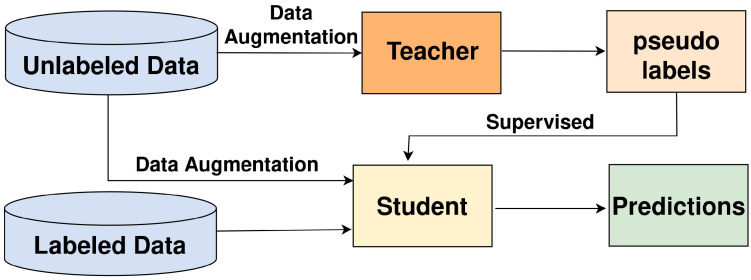
Teacher–Student Architecture for Semi-Supervised Object Detection.

**Figure 3 sensors-26-00310-f003:**
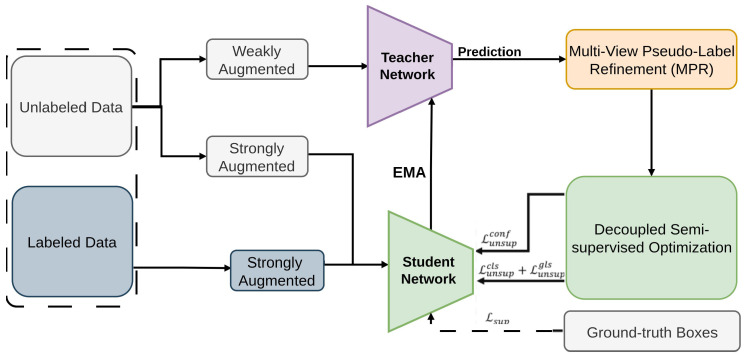
Framework of One Teacher [99]: The teacher generates high-quality pseudo labels for the student via Multi-view Pseudo-label Refinement (MPR) and is updated via EMA. Decoupled Semi-supervised Optimization (DSO) handles multi-task conflicts. YOLOv5 serves as the base model.

**Figure 4 sensors-26-00310-f004:**
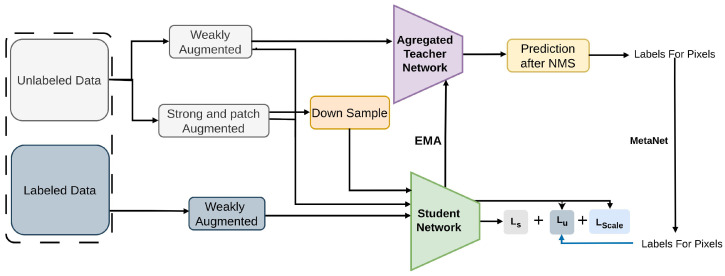
Framework of DSL [95]: The teacher generates pseudo labels for weakly augmented unlabeled images, refined via Adaptive Filtering and MetaNet. Patch-shuffled consistency regularization improves generalization, and the teacher is updated via Aggregated Teacher. The detector is trained with the combined loss.

**Figure 5 sensors-26-00310-f005:**
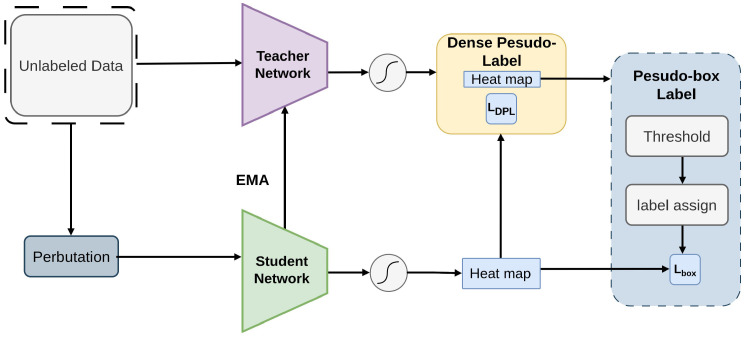
Framework of Dense Teacher [96]: The teacher generates Dense Pseudo-Labels (DPL) for unlabeled images, which guide the student on perturbed inputs. DPL retains rich teacher information, and the total loss combines supervised and unsupervised components.

**Figure 6 sensors-26-00310-f006:**
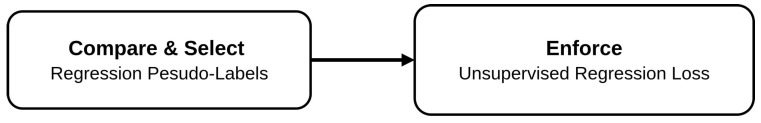
Framework of Unbiased Teacher v2 [96]: Listen2Student improves unsupervised regression by selecting pseudo-labels where the teacher is more confident than the student. Anchor-free detectors show smaller gains from pseudo-labeling compared to anchor-based detectors.

**Figure 7 sensors-26-00310-f007:**
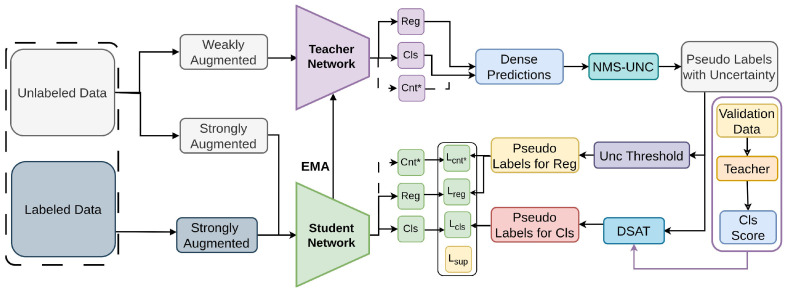
Framework of S4OD [97]: Regression (Reg), Classification (Cls), and Centerness (Cnt*) outputs are used. NMS-UNC generates sparse pseudo labels and retains high-quality regression boxes using an uncertainty threshold. DSAT dynamically selects high-confidence classification boxes based on F1 distribution.

**Figure 8 sensors-26-00310-f008:**
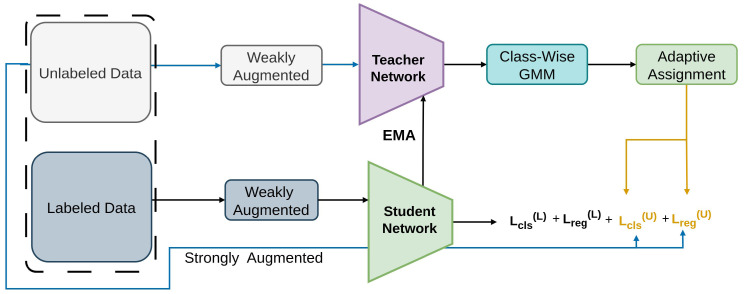
Framework of Consistent Teacher [94]: GMM sets dynamic thresholds, 3D feature alignment improves regression quality, and Adaptive Assignment assigns anchors based on matching cost.

**Figure 9 sensors-26-00310-f009:**
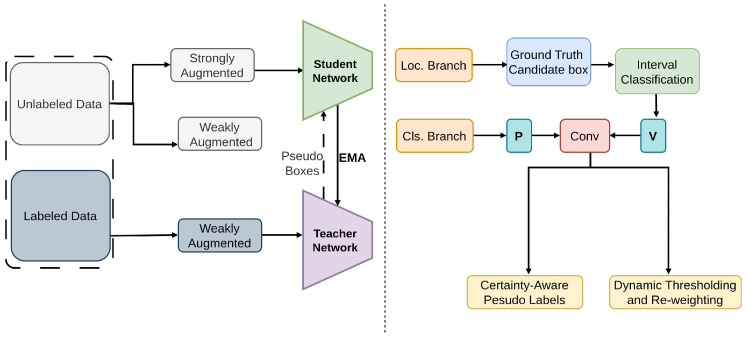
Framework of Rethinking Pse [91]: On the left, the teacher generates pseudo labels from labeled images to train the student. On the right, certainty-aware pseudo labels use classification and localization confidence, with dynamic thresholds and class-wise loss re-weighting to improve localization and mitigate class imbalance.

**Figure 10 sensors-26-00310-f010:**
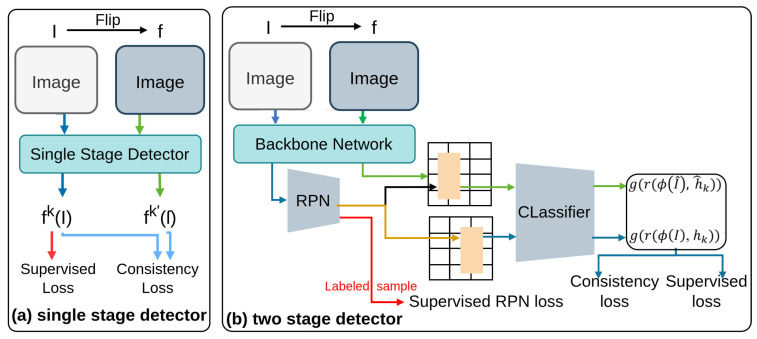
Framework of CSD [77]: (**a**) Single-stage detector applies supervised loss on labeled data and consistency loss between original and flipped features. (**b**) Two-stage detector aligns flipped features with original RoIs, computing supervised and consistency losses similarly. Arrows indicate data or feature flow: blue for supervised loss, green for consistency loss, and red for labeled sample flow.

**Figure 11 sensors-26-00310-f011:**
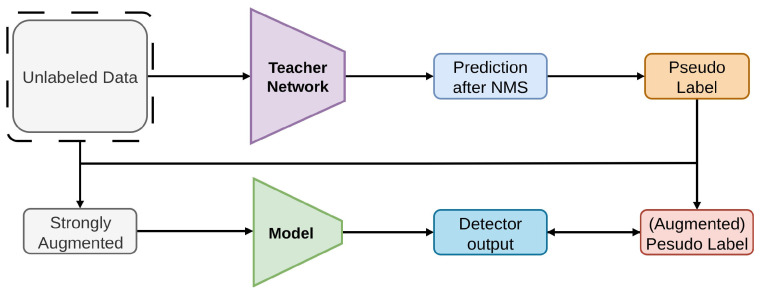
Overview of STAC [78]: Pseudo labels are generated from the teacher model using test-time inference and NMS. Unsupervised loss is applied to high-confidence pseudo labels, with strong augmentations ensuring consistency and target boxes adjusted for global transformations.

**Figure 12 sensors-26-00310-f012:**
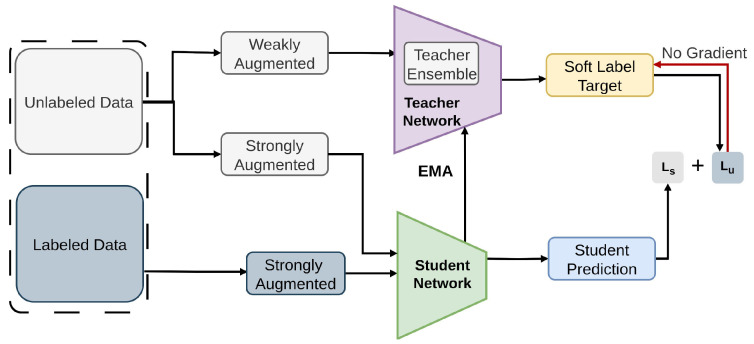
Overview of Humble Teacher [79]: The teacher generates soft pseudo-labels for the student, and its parameters are updated using exponential moving average (EMA).

**Figure 13 sensors-26-00310-f013:**
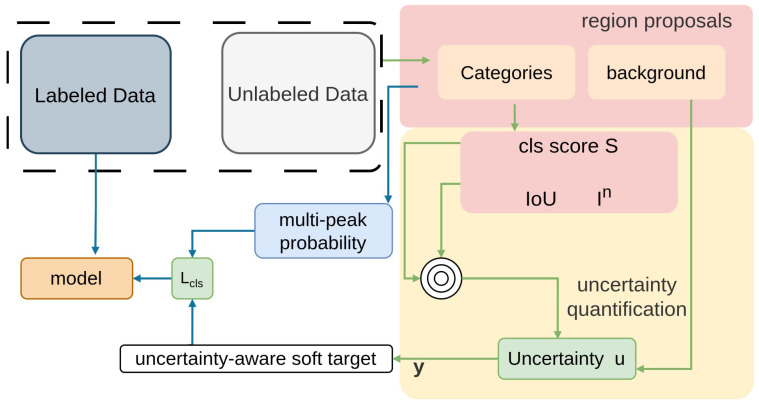
Framework of Combating Noise [84]: Uncertainty is quantified for different regions, and uncertainty-aware soft targets with multi-peak probability distributions are used to incorporate uncertainty into training, enabling noise-resistant learning.

**Figure 14 sensors-26-00310-f014:**
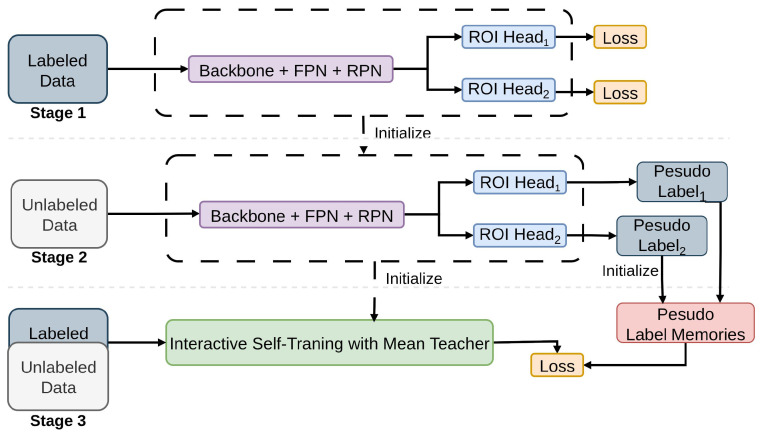
Framework of ISMT [86]: A detection model is first trained on labeled data to generate initial pseudo labels stored in a memory. Semi-supervised training uses the pretrained model with interactive self-training via the mean teacher method. After training, only a single ROI head is needed for inference.

**Figure 15 sensors-26-00310-f015:**
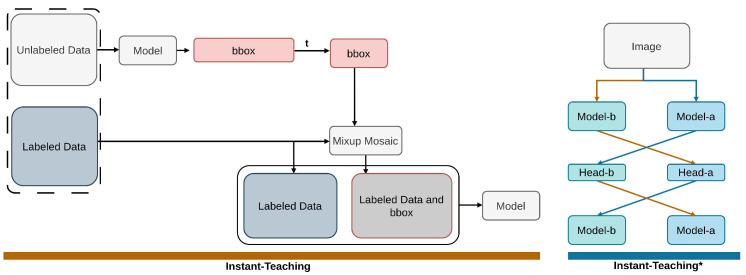
Overview of Instant Teaching [80]: Instant-Teaching applies instant pseudo-labeling along with extended weak and strong augmentations. Instant-Teaching* refers to the approach combined with the co-rectify scheme.

**Figure 16 sensors-26-00310-f016:**
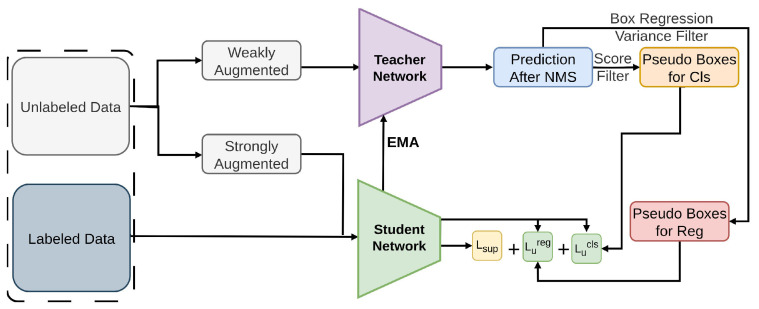
Overview of Soft Teacher [81]: A soft teacher generates pseudo labels on weakly augmented unlabeled images, producing separate sets for classification and regression. The teacher is updated via EMA, and the total loss combines supervised and unsupervised detection losses.

**Figure 17 sensors-26-00310-f017:**
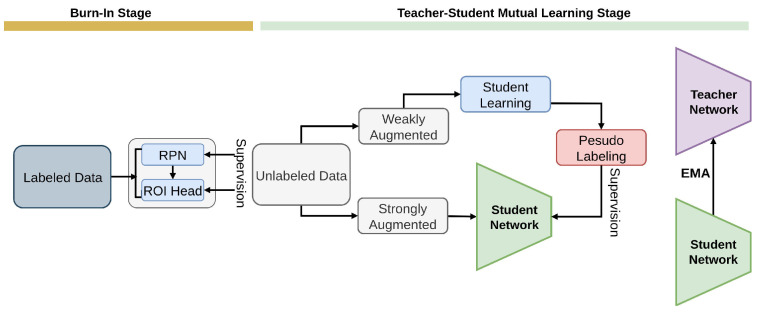
Overview of Unbiased Teacher [82]: Burn-In trains the detector on labeled data. In Teacher–Student Mutual Learning, the fixed teacher generates pseudo-labels for the student, and the student’s knowledge is transferred back to the teacher via EMA. The teacher receives weakly augmented inputs, and the student receives strongly augmented inputs.

**Figure 18 sensors-26-00310-f018:**
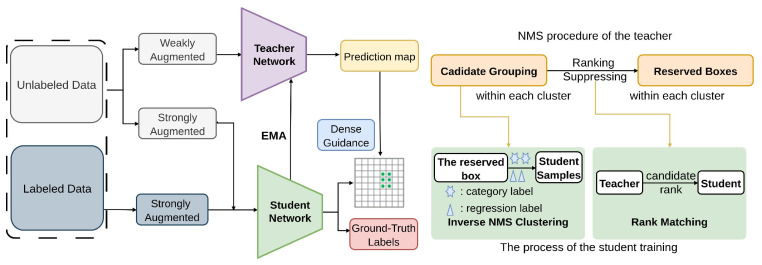
Framework of DTG-SSOD [90]: Training batches contain labeled and unlabeled data. For unlabeled data, the teacher provides dense guidance rather than sparse pseudo labels to supervise the student. The teacher’s NMS procedure guides student training: Inverse NMS Clustering enables the student to group candidates like the teacher, and Rank Matching conveys relational information over the clustered candidates.

**Figure 19 sensors-26-00310-f019:**
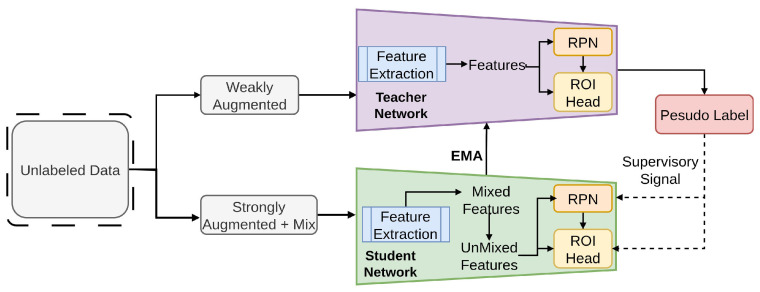
Framework of MUM [85]: The teacher generates pseudo labels to supervise the student. Weakly augmented inputs go to the teacher, and strongly augmented mixed inputs go to the student. Mixed features are unmixed for the student’s detection head, and the teacher is updated via EMA of the student’s weights.

**Figure 20 sensors-26-00310-f020:**
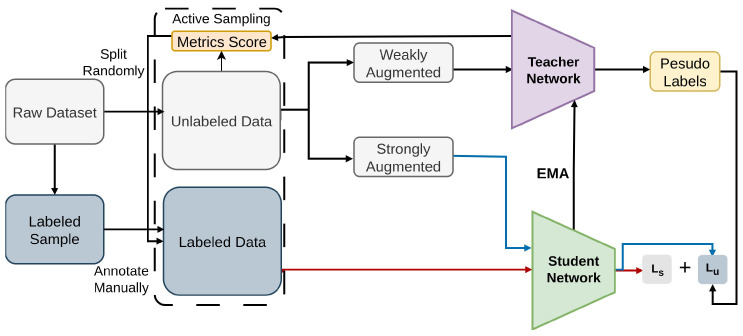
Framework of Active Teacher [92]: Partially initialized labels are gradually expanded. Teacher generates pseudo labels and is updated via EMA, while the Student is trained on both ground-truth and pseudo labels. The Teacher also identifies unlabeled examples for active sampling.

**Figure 21 sensors-26-00310-f021:**
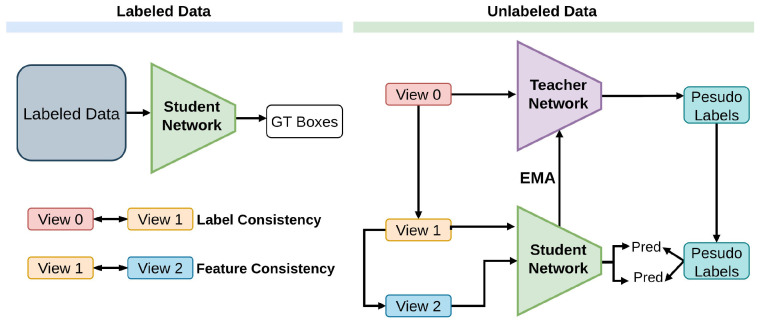
Framework of PseCo [76]: Training batches combine labeled and unlabeled images. The student model is trained on two augmented views (V1 and V2) of the unlabeled data, guided by the same pseudo boxes. The teacher model processes the original input images, denoted as V0.

**Figure 22 sensors-26-00310-f022:**
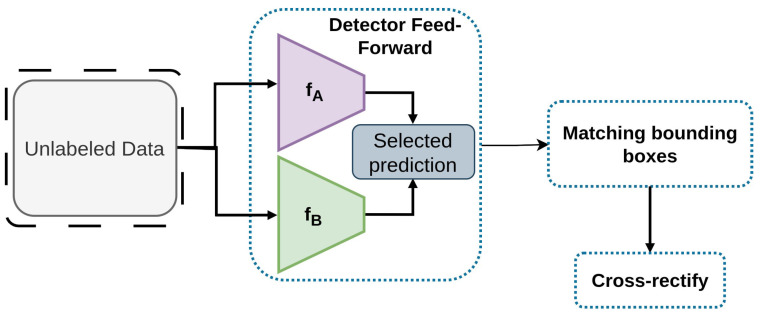
Framework of Cross Rectify [87]: Mechanism of pseudo label generation in the presented framework.

**Figure 23 sensors-26-00310-f023:**
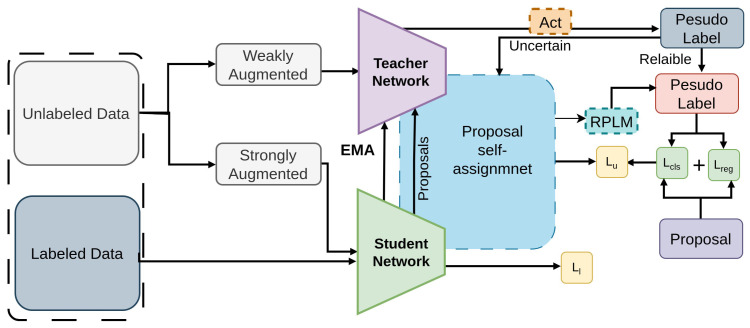
Framework of Label Match [89]: Labeled data train the student with supervised loss. For unlabeled data, the teacher generates pseudo labels using adaptive confidence thresholds (ACT). Reliable labels train the student directly, while uncertain labels are guided by proposal self-assignment. High-quality uncertain labels are gradually upgraded via Reliable Pseudo Label Mining (RPLM).

**Figure 24 sensors-26-00310-f024:**
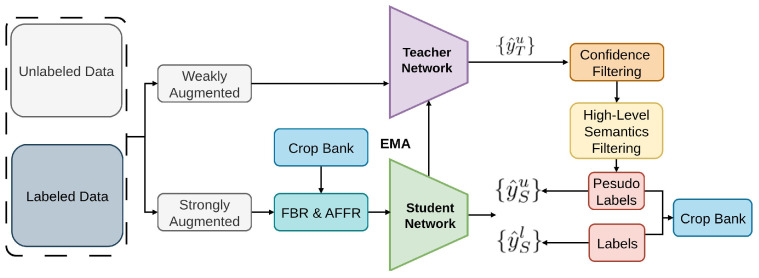
Framework of ACRST [83]: The teacher generates pseudo labels from weakly augmented data, while the student is trained with both ground-truths and pseudo labels. CropBank stores annotations to perform class rebalancing (FBR and AFFR) on strongly augmented data, and a two-stage pseudo-label filtering strategy further improves training.

**Figure 25 sensors-26-00310-f025:**
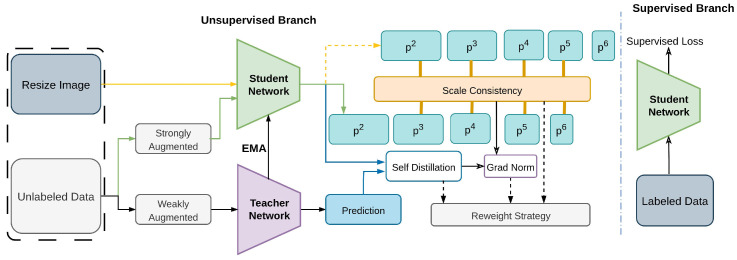
Framework of SED [88]: An FPN-based detector (P2–P6) is used, where the supervised branch shares the student model with the unsupervised branch. ‘sg’ indicates teacher predictions excluded from gradient updates, and Scale Consistency Regularization enforces consistency across feature levels.

**Figure 26 sensors-26-00310-f026:**
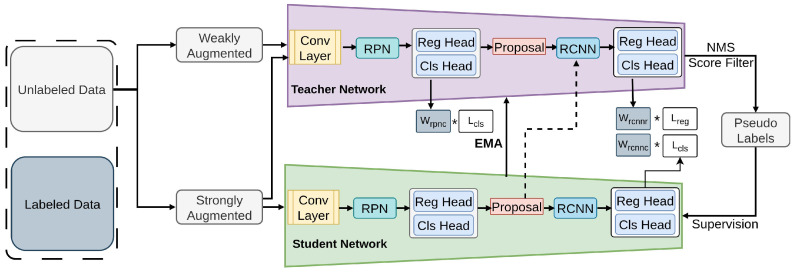
Framework of Self-Correction Mean Teacher [93]: Pseudo-labels are generated from weakly augmented unlabeled data using the teacher model. The unsupervised loss is computed with self-correction weights, indicated by the black dashed line.

**Figure 27 sensors-26-00310-f027:**
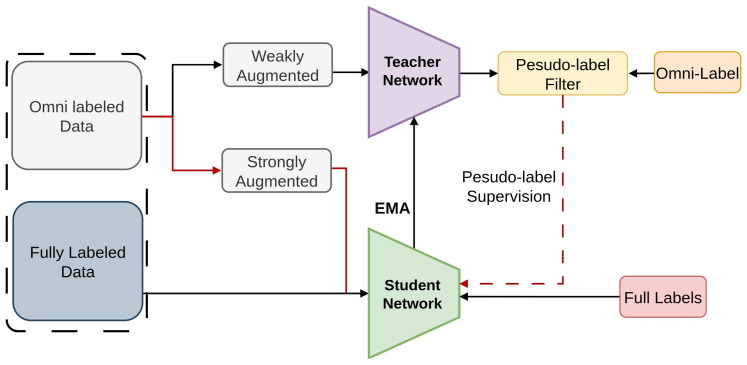
Framework of Omni-DETR [75]: omni-labels filter teacher predictions through a unified pseudo-label filter to generate pseudo-labels for student training.

**Figure 28 sensors-26-00310-f028:**
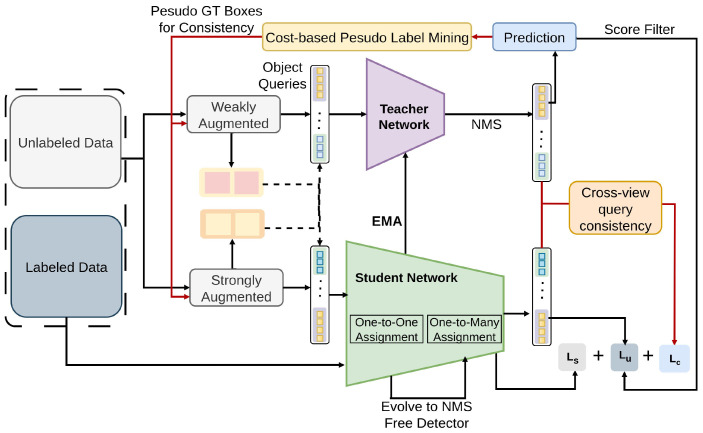
Framework of Semi-Detr [73]: Multi-stage training generates high-quality pseudo labels using Hybrid Matching, followed by one-to-one training. Cross-view query consistency loss with GMM-filtered pseudo boxes enhances overall training consistency.

**Figure 29 sensors-26-00310-f029:**
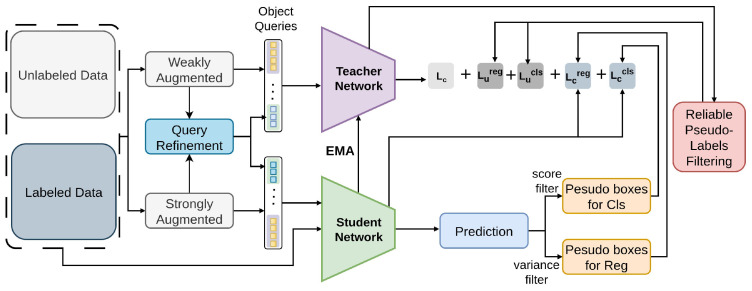
Framework of Sparse Semi-Detr [74]: labeled data trains the student with supervised loss, while the teacher generates pseudo-labels from weakly augmented unlabeled data. Query refinement prevents incorrect matching, and Reliable Pseudo-label Filtering progressively removes low-quality pseudo-labels.

**Figure 30 sensors-26-00310-f030:**
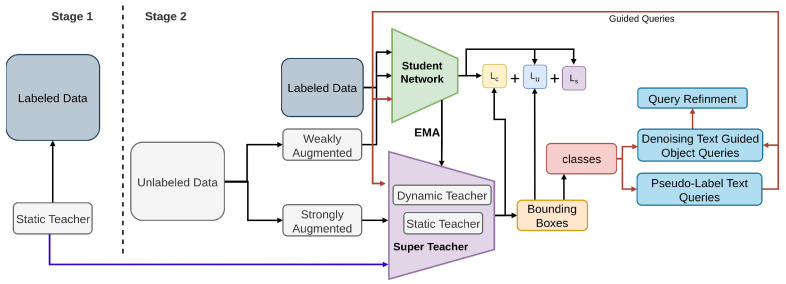
Overview of STEP-DETR [176]. A static teacher is trained on labeled data, while the student learns from both labeled and augmented unlabeled images. The Super Teacher provides high-quality pseudo-labels, aided by text-guided queries and query refinement. Supervised, unsupervised, and consistency losses jointly enhance detection performance.

**Figure 31 sensors-26-00310-f031:**
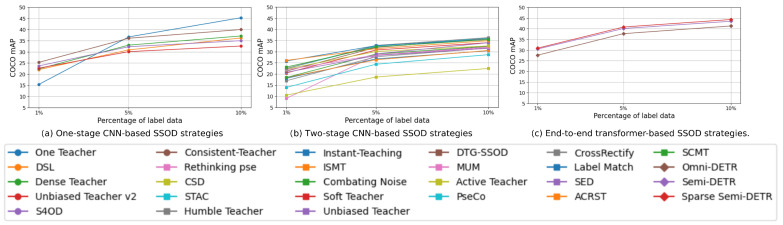
Comparison of CNN-based and transformer-based SSOD strategies on COCO dataset. (**a**) Performance comparison of one-stage CNN-based SSOD Strategies. (**b**) Performance comparison of two-stage CNN-based SSOD Strategies. (**c**) Performance comparison of end to end transformer-based SSOD Strategies.

**Figure 32 sensors-26-00310-f032:**
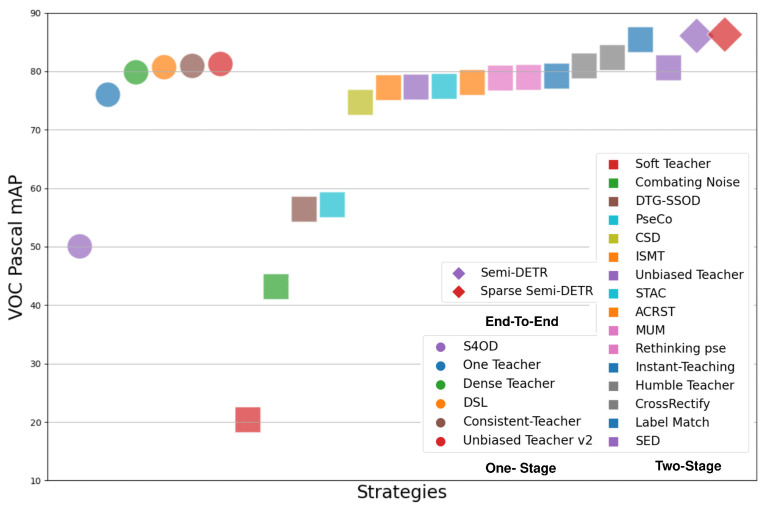
Comparison of CNN-based (one-stage, two-stage) and transformer-based (end-to-end) SSOD strategies on VOC pascal dataset.

**Table 1 sensors-26-00310-t001:** Overview of previous surveys on object detection. For each paper, the publication details are provided.

Title	Year	Description
Semi-Supervised Learning Literature Survey [104]	2008	This survey examines the landscape of semi-supervised learning literature concentrating on diverse methodologies and applications.
A Survey On Semi-Supervised Learning Techniques [108]	2014	An Analysis investigates various techniques in semi-supervised learning, offering insights into their effectiveness and applications.
A Survey and Comparative Study of Tweet Sentiment Analysis via Semi-Supervised Learning [107]	2016	This study provides a thorough comparison and analysis of tweet sentiment methods employing semi-supervised learning techniques.
Semi-supervised learning for medical application: A survey [113]	2018	This paper delves into the integration of semi-supervised learning within medical contexts, offering insights into its applicability and potential advancements.
A survey on semi supervised learning [109]	2019	This comprehensive examination explores the domain of semi-supervised learning, shedding light on its practical implementations and advancements.
Improvability Through Semi-Supervised Learning: A Survey of Theoretical Results [105]	2020	This analysis investigates theoretical advancements facilitated by semi-supervised learning, exploring avenues for improvement within machine learning frameworks.
Small Data Challenges in Big Data Era: A Survey of Recent Progress on Unsupervised and Semi-Supervised Methods [110]	2020	This exploration examines recent progress in unsupervised and semi-supervised methods, addressing challenges posed by small data in the context of the big data era.
A Survey of Un-, Weakly-, and Semi-Supervised Learning Methods for Noisy, Missing and Partial Labels in Industrial Vision Applications [106]	2021	This survey evaluates unsupervised, weakly-supervised, and semi-supervised learning techniques designed to address problems caused by noisy, incomplete, and missing labels in industrial vision applications.
Semi-Supervised and Unsupervised Deep Visual Learning: A Survey [112]	2022	This study explores the field of deep visual learning, with a particular focus on semi-supervised and unsupervised methods. It aims to uncover key insights and advancements in these approaches.
A Survey on Semi-, Self- and Unsupervised Learning for Image Classification [111]	2022	This survey examines image classification, focusing on semi-supervised, self-supervised, and unsupervised learning methods to understand their effectiveness and potential applications.
A survey on semi-supervised learning for delayed partially labelled data streams [114]	2022	This study delves into semi-supervised learning approaches employed for handling delayed data streams with semi labels, focusing on their effectiveness and challenges.
Semi Supervised deep learning for image classification with distribution mismatch: A survey [115]	2022	This study explores Semi-Supervised Deep Learning for image classification with distribution mismatch, providing insights into its strategies and challenges.
A Survey on Deep Semi-supervised Learning [49]	2023	This survey examines the field of deep semi-supervised learning techniques, providing insights into their applications and advancements.
Graph-based semi-supervised learning: A comprehensive review [116]	2023	This comprehensive review examines the effectiveness and applications of graph-based semi-supervised learning methods.

**Table 2 sensors-26-00310-t002:** Object Detection Performance on COCO-Partial Dataset. Comparison of object detection methods across different stages on the COCO-Partial dataset.

Methods	Stages	Reference	COCO-Partial		
			1%	5%	10%
One Teacher [99]	One Stage	-	15.4	36.70	45.3
DSL [95]		CVPR22	22.03	30.87	36.22
Dense Teacher [96]		ECCV22	22.38	33.01	37.13
Unbiased Teacher v2 [98]		CVPR22	22.71	30.08	32.61
S4OD [97]		-	23.70	32.30	35.00
Consistent-Teacher [94]		CVPR23	25.30	36.10	40.00
Rethinking pse [91]	Two Stage	AAAI22	9.02	28.40	32.23
CSD [77]		ICML23	10.51	18.63	22.46
STAC [78]		-	13.97	24.38	28.64
Humble Teacher [79]		CVPR22	16.96	27.70	31.61
Instant-Teaching [80]		CVPR21	18.05	26.75	30.40
ISMT [86]		CVPR21	18.41	26.37	30.53
Combating Noise [84]		-	18.41	28.96	32.43
Soft Teacher [81]		ICCV21	20.46	30.74	34.04
Unbiased Teacher [82]		ICLR21	20.75	28.27	31.50
DTG-SSOD [90]		-	21.27	31.90	35.92
MUM [85]		CVPR22	21.88	28.52	31.87
Active Teacher [92]		CVPR22	22.20	30.07	32.58
PseCo [76]		ECCV22	22.43	32.50	36.06
CrossRectify [87]		CVPR22	22.50	32.80	36.30
Label Match [89]		CVPR22	25.81	32.70	35.49
ACRST [83]		-	26.07	31.35	34.92
SED [88]		CVPR22	-	29.01	34.02
SCMT [93]		IJCAI22	23.09	32.14	35.42
Omni-DETR [75]	End to End	CVPR22	27.60	37.70	41.30
Semi-DETR [73]		CVPR23	30.50	40.10	43.5
Sparse Semi-DETR [74]		CVPR24	30.90	40.80	44.30
STEP DETR [176]		ICCV25	31.70	41.1	45.4

**Table 3 sensors-26-00310-t003:** Object Detection Performance on PASCAL-VOC Dataset. Comparison of object detection methods across different stages on the PASCAL-VOC dataset.

Methods	Stages	Reference	PASCAL-VOC		
			${\mathrm{AP}}_{50}$	mAP	${\mathrm{AP}}_{75}$
S4OD [97]	One Stage	-	50.1	-	34.0
Dense Teacher [96]		ECCV22	79.89	55.87	-
DSL [95]		CVPR22	80.7	56.8	-
Consistent-Teacher [94]		CVPR23	81.00	59.00	-
Unbiased Teacher v2 [98]		CVPR22	81.29	56.87	-
One Teacher [99]		-	76.1	-	-
Soft Teacher [81]	Two Stage	ICCV21	20.46	30.74	34.04
Combating Noise [84]		-	43.2	62.0	47.5
DTG-SSOD [90]		-	56.4	-	38.8
PseCo [76]		ECCV22	57.2	-	39.2
CSD [77]		ICML23	74.70	-	-
ISMT [86]		CVPR21	77.23	46.23	-
Unbiased Teacher [82]		ICLR21	77.37	48.69	-
STAC [78]		-	77.45	44.64	-
ACRST [83]		-	78.16	50.1	-
MUM [85]		CVPR22	78.94	50.22	-
Rethinking pse [91]		AAAI22	79.0	54.60	59.4
Instant-Teaching [80]		CVPR21	79.20	50.00	54.00
Humble Teacher [79]		CVPR22	80.94	53.04	-
CrossRectify [87]		CVPR22	82.34	-	-
Label Match [89]		CVPR22	85.48	55.11	-
SED [88]		CVPR22	80.60	-	-
Semi-DETR [73]	End to End	CVPR23	86.10	65.2	-
Sparse Semi-DETR [74]		CVPR24	86.30	65.51	-
STEP DETR [176]		ICCV25	86.85	65.87	-

**Table 4 sensors-26-00310-t004:** A brief description of Advantages and limitations of Semi Supervised Strategies.

Methods	Advantages	Limitations
Stac [78]	Improves detection performance with minimal complexity.	Low performance with frameworks employing intense hard negative mining, leading to over dependence on noisy pseudo-labels.
Humble Teacher [79]	Improves performance significantly with dynamic teacher model updates and soft pseudo-labels.	More computational resources due to the dynamic updating of the teacher model and the ensemble of numerous teacher models, potentially increasing training time and complexity.
Instant Teaching [80]	Improving model learning with extended weak-strong data augmentation as well as instant pseudo labeling.	Dependency on Extensive weak-strong data augmentation and instant pseudo labeling introduce computational overhead, increase training complexity and time.
Soft Teacher [81]	Enhances detector performance and pseudo label quality simultaneously.	Depending on extensive data augmentation and the soft teacher approach potentially increase training complexity and computational overhead.
Unbiased Teacher [82]	Effectively mitigates pseudo-labeling bias and overfitting in Semi-Supervised Object Detection.	Relies on the balance between the student and teacher models, which require careful tuning and additional computational resources.
ACRST [83]	Improves performance by addressing class imbalance.	Effectiveness relies on the precision of pseudo-labels, which are impacted by noise due to the complexity of detection tasks, requiring robust filtering mechanism.
Combating Noise [84]	Effectively combating noise associated with pseudo labels enhances the robustness of the SSOD Tasks.	Dependence on accurately quantifying region uncertainty is challenging in complex scenes or datasets with diverse object characteristics.
MUM [85]	Effectively augments data for Semi-Supervised Object Detection, enhancing model robustness without significant computational overhead.	Encounter difficulties in accurately locating object boundaries due to the mixing process, potentially affecting localization precision.
ISTM [86]	Effectively leveraging ensemble learning to enhance the usefulness of pseudo labels and stabilize Semi-Supervised Object Detection training.	Introduce additional computational complexity due to the ensemble approach and the use of multiple ROI heads, potentially increasing training time and resource requirements.
Cross Rectify [87]	Enhances pseudo label quality and detection performance by rectifying misclassified bounding boxes using detector disagreements.	Simultaneous training of two detectors increase computational overhead, potentially prolonging training time and resource usage.
SED [88]	Improves Semi-Supervised Object Detection by enforcing scale consistency and self-distillation.	Reliance on the IoU threshold criterion, which could not be optimal for all detectors and situations, and its limited benefits from multi-scale testing
Label Match [89]	Improves Semi-Supervised Object Detection by addressing label mismatch through distribution-level and instance-level methods.	Assumes Both unlabeled as well as labeled data have the same distribution, potentially restricting its applicability in diverse scenarios.
DTG-SSOD [90]	Leverages Dense Teacher Guidance for more accurate supervision, enhancing Semi-Supervised Object Detection performance.	Implementation complexity, especially with Inverse NMS Clustering and Rank Matching, increase computational resources during training.
Rethinking Pse [91]	Certainty-aware pseudo labels improve performance by addressing localization precision and class imbalance issues	Implementing certainty-aware pseudo labeling add additional computational complexity during training.
CSD [77]	Leverages consistency constraints for both classification and localization, enhancing object detection performance using unlabeled data.	It shows less performance improvement in two-stage detectors compared to single-stage detectors.
PseCo [76]	Enhances SSOD by integrating object detection attributes into pseudo labeling along with consistency training, leading to superior performance and faster convergence.	Its potential struggle with generalization across diverse datasets due to variability in pseudo-label quality.
Active Teacher [92]	Maximizes limited label information through active sampling, enhancing pseudo-label quality and improving SSOD performance.	Require more training steps compared to other methods, potentially increasing computational overhead.
One Teacher [99]	Improves SSOD on YOLOv5, tackling issues like low-quality pseudo-labeling.	Lowering the threshold for pseudo-labeling due to noisy pseudo-labeling in one-stage detection makes it difficult to maximize the effectiveness of one-stage teacher–student learning.
Dense Teacher [96]	Simplifies the SSOD pipeline by using Dense Pseudo-Labels, improving efficiency and performance.	Contain high-level noise, potentially impacting detection performance if not properly addressed.
Unbiased Teacher v2 [82]	Expands the applicability of SSOD to anchor-free detectors, improving performance across various benchmarks.	Challenges remain in scaling the method to large datasets, integrating localization uncertainty estimation for boundary prediction with the relative thresholding mechanism, and addressing domain shift issues.
S4OD [97]	Dynamically adjusts pseudo-label selection to balance quality and quantity, enhancing single-stage detector performance	DSAT strategy’s increased time cost is due to F1-score computation, and using the CPU version of NMS for uncertainty computation slows down training.
Consistent-Teacher [94]	Improves SSOD performance by addressing inconsistent pseudo-targets with feature alignment, adaptive anchor assignment, and dynamic threshold adjustment.	Performance is validated mainly on single-stage detectors, with effectiveness on stage-two detectors and DETR-based models yet to be confirmed.
Omni-DETR [75]	utilize diverse weak annotations to enhance performance and annotation efficiency.	Effectiveness on larger datasets is uncertain, and its simplified annotation process could raise concerns about potential misuse.
Semi-DETR [73]	Combines Cross-view query consistency and stage-wise hybrid matching to improve training efficiency.	Encounter challenges due to the absence of deterministic connection between the predictions and the input queries.
Sparse Semi-DETR [74]	Introduces a Query Refinement Module to improve object query functionality, enhancing detection performance for small and obscured objects.	Require additional computational resources due to the integration of novel modules, potentially increasing training time and complexity.

## Data Availability

The original contributions presented in this study are included in the article. Further inquiries can be directed to the corresponding authors.

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
