# Peer review of "Semi-Supervised Object Detection: A Survey on Progress from CNN to Transformer"

_sensors, 2026, doi:10.3390/s26010310_

Round 1
Reviewer 1 Report
Comments and Suggestions for Authors
The article lacks sufficient summarization and conciseness regarding algorithms, with a notable absence of articles from the past two years. Additionally, the experimental results for various algorithms presented in the article lack persuasiveness.
Comments on the Quality of English LanguageEnglish is sufficient to convey the main idea of the article
Author Response
Reviewer 1
The article lacks sufficient summarization and conciseness regarding algorithms, with a notable absence of articles from the past two years. Additionally, the experimental results for various algorithms presented in the article lack persuasiveness.
Thank you for pointing this out. We have revised Section 2 by adding recent strategies from the past two years. Furthermore, the experimental results presented for the various algorithms are drawn directly from the original papers to ensure accuracy.
Reviewer 2 Report
Comments and Suggestions for Authors
Dear Authors,
This survey tackles Semi-Supervised Object Detection (SSOD), a highly relevant research area given the cost and scarcity of labeled data in real-life applications. You have selected a significant number of methods and references, which is commendable. However, the current presentation lacks the critical evaluation, deep comparison, and rigorous analysis expected of a survey paper.
Following are some directions to refine the manuscript and strengthen the contribution.
-
Introduction: This has an appropriate length and introduces the problem in depth.
-
Related previous reviews and surveys: This section is not necessary. It doesn't make any meaningful contribution to the paper. I suggest you remove it or combine it with Section 3.
-
Related work: I'm not sure whether this section topic is appropriate, as this is a survey paper. Reconsider the title of the subtopic. Also, I would prefer to see a detailed discussion on how you selected your references/papers for this survey. What is your criterion? When presenting key contributions and methodologies in SSOD, I advise the authors to analyze each technique much deeper than simply stating the method. Discuss the shortcomings of each technique presented.
-
Semi-supervised strategies: In this section, you have presented many strategies, which is good. Yet, you don't present any detailed analysis, such as the implementation approach, bottlenecks, advantages, and disadvantages of each method. The diagrams are helpful for comparison, though.
-
Loss functions: Again, you are just listing each and every function. There is no mention of which models prefer these functions and how they impact SSOD.
-
Datasets and comparison: Figures 30 and 31 present some comparison study. Yet, you don't discuss how to use labeled datasets such as MS-COCO for SSOD. Section 6.1 fails to add any significant contribution. However, Section 6.2, with Figures 30 and 31, does carry a noteworthy analysis.
-
Open Challenges and Future directions: Advice to improve the discussion here. Comment about the system complexity, challenges in deploying such systems, and methods adopted to maintain the accuracy of the detection.
-
Application: You need to go beyond merely stating applications. Discuss the specific challenges SSOD solves in these applications and the unique architectural requirements for successful deployment.
other general comments:
-
The researchers have gone through a significant number of papers. I suggest you reformat the paper's presentation and add critical discussion to the manuscript where needed. You need to go beyond simply stating the facts and provide deeper analysis.
-
Rearrange the sub-sections in the paper. Sections 3, 4, and 5, in particular, have a significantly large number of subtopics. Please reconsider this arrangement and introduce a few logical groupings (e.g., categories or classification types).
-
The language and writing of the paper are generally acceptable. Kindly add deeper analysis to make the manuscript more appealing and valuable to the readers.
Author Response
Reviewer 2
This survey tackles Semi-Supervised Object Detection (SSOD), a highly relevant research area given the cost and scarcity of labeled data in real-life applications. You have selected a significant number of methods and references, which is commendable. However, the current presentation lacks the critical evaluation, deep comparison, and rigorous analysis expected of a survey paper.
Following are some directions to refine the manuscript and strengthen the contribution.
- Introduction: This has an appropriate length and introduces the problem in depth.
Thank you for your comment. The Introduction has been revised to provide a detailed and well-structured overview of Semi-Supervised Object Detection, covering the motivation, problem statement, evolution from CNN to Transformer-based approaches, key methodologies, and a clear roadmap of the paper.
- Related previous reviews and surveys: This section is not necessary. It doesn't make any meaningful contribution to the paper. I suggest you remove it or combine it with Section 3.
This section has been removed as suggested. The relevant material has been merged into the Introduction to avoid redundancy and improve the flow.
- Related work: I'm not sure whether this section topic is appropriate, as this is a survey paper. Reconsider the title of the subtopic. Also, I would prefer to see a detailed discussion on how you selected your references/papers for this survey. What is your criterion? When presenting key contributions and methodologies in SSOD, I advise the authors to analyze each technique much deeper than simply stating the method. Discuss the shortcomings of each technique presented.
It is in comparison and discussion section
Thank you for the suggestion. We have removed the Related Work section as recommended. Additionally, we have added a detailed explanation in Section 2 describing the criteria used for selecting the papers included in this survey. The key contributions, methodologies, and shortcomings of the reviewed SSOD techniques are now more thoroughly analyzed in the 5th section. - Semi-supervised strategies: In this section, you have presented many strategies, which is good. Yet, you don't present any detailed analysis, such as the implementation approach, bottlenecks, advantages, and disadvantages of each method. The diagrams are helpful for comparison, though.
Thank you for the suggestion. While the Semi-supervised Strategies section provides an overview of the main approaches, the detailed analysis is now explained in Sections 4 and 5, where we provide a deeper comparative discussion. - Loss functions: Again, you are just listing each and every function. There is no mention of which models prefer these functions and how they impact SSOD.
Thank you for pointing this out. We have revised the Loss Functions section. - Datasets and comparison: Figures 30 and 31 present some comparison study. Yet, you don't discuss how to use labeled datasets such as MS-COCO for SSOD. Section 6.1 fails to add any significant contribution. However, Section 6.2, with Figures 30 and 31, does carry a noteworthy analysis.
Thank you for your feedback. We have revised Section Datasets and Comparison to explain the use of labeled datasets in SSOD and provide a detailed comparison showing the advantages of transformer-based methods over CNN-based approaches. - Open Challenges and Future directions: Advice to improve the discussion here. Comment about the system complexity, challenges in deploying such systems, and methods adopted to maintain the accuracy of the detection.
Thank you for your feedback. We have revised the Open Challenges & Future Directions section to discuss system complexity, deployment challenges, and strategies for maintaining detection accuracy. - Application: You need to go beyond merely stating applications. Discuss the specific challenges SSOD solves in these applications and the unique architectural requirements for successful deployment.
Thank you for your feedback. We have revised the Applications section to go beyond listing use cases by discussing the specific challenges SSOD addresses in each domain and the architectural requirements for effective deployment.
other general comments:
- The researchers have gone through a significant number of papers. I suggest you reformat the paper's presentation and add critical discussion to the manuscript where needed. You need to go beyond simply stating the facts and provide deeper analysis.
- Rearrange the sub-sections in the paper. Sections 3, 4, and 5, in particular, have a significantly large number of subtopics. Please reconsider this arrangement and introduce a few logical groupings (e.g., categories or classification types).
- The language and writing of the paper are generally acceptable. Kindly add deeper analysis to make the manuscript more appealing and valuable to the readers.
Thank you for your feedback. We have reorganized the sub-sections for better logical grouping and added deeper analysis and critical discussion throughout the manuscript to enhance clarity and value.
Reviewer 3 Report
Comments and Suggestions for Authors
Please see the attachment for detailed comments.

Author Response
Reviewer 3
This survey provides a comprehensive overview of the evolution of Semi-Supervised Object
Detection (SSOD) from CNN-based to Transformer-based frameworks, covering key aspects such as
pseudo-labeling, teacher–student architectures, loss functions, and data augmentation strategies.
While the paper demonstrates strong coverage and academic potential, it still requires refinement in
visual presentation, methodological comparison, experimental transparency, and discussion depth.
The detailed comments are as follows:
- Figure 3 partially duplicates content from the referenced papers. It is recommended to redraw
or standardize the citation format. Similar issues occur in Figures 4, 7, and 8. As a review
paper, proper citation and image copyright compliance are essential; the authors should
recheck all figures throughout the manuscript.
Thank you for pointing this out. We have carefully reviewed all figures throughout the manuscript and ensured that any content previously duplicating referenced papers has been corrected.
- In Section 3 (“Related Work”), the explanations are purely textual and somewhat abstract.
Adding visual diagrams or framework illustrations to depict the key workflows of
representative methods would significantly improve readability.
Thank you for pointing this out. As suggested by other reviewers, we have removed the Related Work section entirely.
- In Section 4, most method descriptions follow a repetitive “architecture + reference + brief
comment” pattern. The authors should include a systematic comparison among methods—
perhaps through a summary paragraph or a table highlighting their strengths, limitations, and
use cases.
Thank you for your comment. A systematic comparison of the methods, highlighting their strengths, limitations has been added in Section 4, 5.
- The “Loss Functions” section only provides a general description of each function without
explaining its application logic. It is recommended to include the mathematical formulations
and clarify the specific role of each loss in SSOD.
Thank you for pointing this out. We have revised the Loss Functions section to include mathematical formulations and clarify each loss’s role in SSOD.
- Tables 2 and 3 lack information about the source of experimental results. Please specify
whether the results are reproduced or taken directly from the cited works. If reproduced,
details about the experimental environment and hyperparameters should be provided for
fairness and reproducibility.
Thank you for the comment. The results presented in Tables 2 and 3 are taken directly from the cited works, ensuring accuracy and consistency with the original publications.
- Section 8’s discussion on application scenarios remains too broad. The authors should expand
on specific real-world case studies or representative papers to demonstrate the practical value
of SSOD methods.
Thank you for your feedback. We have revised this Section to include specific real-world case studies and representative papers, demonstrating the practical value of SSOD methods.
- The conclusion section should emphasize the main contributions of this survey, especially its
novelty in bridging CNN-based and Transformer-based SSOD approaches and offering a
structured comparative perspective.
Thank you for your suggestion. We have revised the Conclusion section to clearly emphasize the main contributions of this survey, highlighting its novelty in bridging CNN-based and Transformer-based SSOD approaches and providing a structured comparative perspective.
- The reference numbering and in-text citation order need verification.
Thank you for pointing this out. We have carefully reviewed and verified all reference numbering and in-text citations
Round 2
Reviewer 1 Report
Comments and Suggestions for Authors
Accept in present form
Comments on the Quality of English LanguageEnglish is sufficient to convey the main idea of the article
Author Response
We thank the reviewer for the positive evaluation and the recommendation to accept the manuscript in its present form. We acknowledge the comment regarding English clarity and have revised the manuscript to improve language, readability, and expression while maintaining the original meaning. We appreciate the reviewer’s valuable feedback.
Reviewer 2 Report
Comments and Suggestions for Authors
Dear Authors,
Most comments are addressed in the revised version.
The following are further comments to improve the manuscript.
- In Section 2, you should discuss the training methodology for each model presented. Since you have included more than twenty models, the discussion must specify which dataset and loss functions are suitable for each model, and also present a comparison of the application scenarios for every model.
- Please note the following formatting guideline: Figure captions should be placed below the figure, while Table captions must be placed at the top of the table.
- Kindly ensure the discussion moves beyond the simple presentation of facts. Only by doing so will you add the necessary depth to the paper and invite the reader to continue reading.
Author Response
Dear Authors,
Most comments are addressed in the revised version.
The following are further comments to improve the manuscript.
- In Section 2, you should discuss the training methodology for each model presented. Since you have included more than twenty models, the discussion must specify which dataset and loss functions are suitable for each model, and also present a comparison of the application scenarios for every model.
Thank you for your response. Section 2 has been revised to clearly describe the training methodology of each model, including suitable datasets, loss functions, and a comparative discussion of their application scenarios.
- Please note the following formatting guideline: Figure captions should be placed below the figure, while Table captions must be placed at the top of the table.
We have corrected the manuscript formatting so that all figure captions are placed below the figures and all table captions are positioned at the top of the tables.
- Kindly ensure the discussion moves beyond the simple presentation of facts. Only by doing so will you add the necessary depth to the paper and invite the reader to continue reading.
We have expanded the discussion to go beyond a simple presentation of facts by adding critical analysis, highlighting strengths and limitations, and discussing practical implications.